# The Quadrature Method: A Novel Dipole Localisation Algorithm for Artificial Lateral Lines Compared to State of the Art

**DOI:** 10.3390/s21134558

**Published:** 2021-07-02

**Authors:** Daniël M. Bot, Ben J. Wolf, Sietse M. van Netten

**Affiliations:** 1I-BioStat, Data Science Institute, Hasselt University, 3500 Hasselt, Belgium; 2Delft Center for Systems and Control, Delft University of Technology, 2628 CD Delft, The Netherlands; b.j.wolf@rug.nl; 3Bernoulli Institute of Mathematics, Computer Science and Artificial Intelligence, Faculty of Science and Engineering, University of Groningen, 9747 AG Groningen, The Netherlands

**Keywords:** hydrodynamic imaging, dipole localisation, artificial lateral line, neural networks

## Abstract

The lateral line organ of fish has inspired engineers to develop flow sensor arrays—dubbed artificial lateral lines (ALLs)—capable of detecting near-field hydrodynamic events for obstacle avoidance and object detection. In this paper, we present a comprehensive review and comparison of ten localisation algorithms for ALLs. Differences in the studied domain, sensor sensitivity axes, and available data prevent a fair comparison between these algorithms from their original works. We compare them with our novel quadrature method (QM), which is based on a geometric property specific to 2D-sensitive ALLs. We show how the area in which each algorithm can accurately determine the position and orientation of a simulated dipole source is affected by (1) the amount of training and optimisation data, and (2) the sensitivity axes of the sensors. Overall, we find that each algorithm benefits from 2D-sensitive sensors, with alternating sensitivity axes as the second-best configuration. From the machine learning approaches, an MLP required an impractically large training set to approach the optimisation-based algorithms’ performance. Regardless of the data set size, QM performs best with both a large area for accurate predictions and a small tail of large errors.

## 1. Introduction

Artificial lateral lines (ALLs) are sensor arrays inspired by the biological lateral line organ found in fish and amphibians. This organ enables fish to detect and locate moving objects such as prey, predators, or social partners [1]. The ability to sense one’s environment using a lateral line is sometimes called hydrodynamic imaging [2,3]. Two types of hydrodynamic imaging are distinguished: active hydrodynamic imaging, where fish use their movement’s flow field to detect stationary obstacles; and passive hydrodynamic imaging, where fish detect fluid flows generated externally. Both types of hydrodynamic imaging have applications for ALLs. Active hydrodynamic imaging is useful for obstacle avoidance of autonomous underwater vehicle (AUVs) [4]. Here, ALLs provide a complementary sense for AUVs because—unlike cameras—they do not rely on visibility and—unlike sonar—they do not actively emit a signal. Passive hydrodynamic imaging can be used for tracking the location of objects—for instance, ships in a harbour—or detecting disturbances near underwater installations.

Several dipole localisation algorithms have been developed for ALLs in the last 15 years [5,6,7,8,9,10,11,12,13,14]. These algorithms attempt to locate objects that move underwater using the water flow pattern their movement generates. This process is analogous to solving the inverse problem of hydrodynamic source localisation [15]. It is challenging to compare these algorithms’ performance from their original works due to differences in experimental designs and conditions. For instance, some algorithms were evaluated for 2D localisation [5,6,7,8,9,10], whereas others localised sources in 3D [11,12,13]. Additionally, some algorithms were evaluated in simulations [5,9], while other studies used physical sensors [6,7,8,10,11,12,13,14]. In particular, differences in the (simulated) sensor sensitivity and the areas in which sources are located prevent the comparison of average or median prediction errors between studies because the number of inaccurate predictions depends heavily on sensor sensitivity and increases with the area’s size.

The present research compares ten dipole localisation algorithms via their ability to determine an object’s state from velocity measurements simulated using potential flow [16]. In our study, this state refers to the 2D position and movement direction relative to an array of flow sensors. The quality of these state estimates is compared using two analyses. First, we determine the size of the area in front of the array in which these algorithms accurately estimate an object’s state. This analysis allows an intuitive comparison of the effective range of each algorithm. Secondly, we determine each algorithm’s distribution of prediction errors, i.e., the differences between the actual and estimated state. This analysis better indicates the reliability of each algorithm and provides a median error metric.

As our baseline, we use two localisation methods: a random predictor (RND) and an off-the-shelf least square curve fit (LSQ) algorithm. The remaining algorithms are divided into three categories. The first category contains three template-based algorithms. Template matching [6] and linear constraint minimum variance (LCMV) beamforming [11,12] use a set of velocity measurements of a single source at different known positions and movement directions to locate a source. The continuous wavelet transform (CWT)—introduced for dipole localisation by Ćurčić-Blake and van Netten [5]—is also a template-based algorithm. This algorithm is based on the observation that a set of wavelets fully describes the potential flow [16] velocity generated by a dipole source. The second category contains artificial neural networks. A multi-layer perceptron (MLP) was used by Abdulsadda and Tan [7] and Boulogne et al. [9]. The latter showed that an extreme learning machine (ELM) performed better than their MLP implementation in high signal-to-noise conditions. The third category contains model-based algorithms. The first two, GN and Newton–Raphson (NR), fit a potential flow model to measured velocity values to predict a dipole’s position and movement direction [8]. Abdulsadda and Tan [8] showed that GN consistently performed slightly better than NR, and both algorithms outperformed LCMV beamforming.

There are several novel aspects of the present study. Firstly, all localisation algorithms are extended to use various combinations of the velocity field’s parallel and perpendicular components with respect to the sensor array. Several 2D sensitive fluid flow sensors exist in the literature [17,18,19,20], which provide the orthogonal fluid flow component that is not yet used by most dipole localisation algorithms. In addition, hair cells with varying orientations in close proximity have been observed in the ear of fish [21] and on the body of the *Xenopus laevis* frog (as cited in [1]). These varying orientations are thought to contribute to the localisation of stimuli. Secondly, not only are both the position and direction of movement of the source varied, their mutual effects on the prediction error are analysed as well. Thirdly, we use a novel approach to compare the performance of the dipole localisation algorithms, quantified by the area in which they correctly predict the position and direction of movement of an object with a predefined accuracy. Fourthly, we extend the template matching algorithm to a k-nearest neighbours (KNN) generalisation, where k=1 is equivalent to the template matching algorithm as referenced earlier [6]. Finally, we introduce a novel model-based dipole localisation algorithm coined the quadrature method (QM), which exploits geometric properties of a 2D-projection of a velocity field. We show that the QM has state-of-the art localisation performance and how the movement direction of a dipole can be estimated directly from velocity measurements and its estimated location.

The remainder of the present paper is structured as follows: Section 2 explains the fluid flow simulation, our methods of analysis, and the dipole localisation algorithms. Section 3 presents the results of both analysis methods, providing error distributions of the algorithms as well as the total area with median errors below predefined levels of accuracy. Section 4 places our findings in the context of previous work. Section 5 summarises our contributions and conclusions.

## 2. Materials and Methods

In the following subsections, we describe the dipole flow field, the simulation environment, the conditions used for our analyses, each dipole localisation algorithm, and their hyperparameter optimisation strategy.

### 2.1. The Dipole Flow Field

Fluid flows were computed for a small sphere (radius a=1 cm) vibrating with a fraction of its size (amplitude A=2 mm), which generates a dipole field [16]. The dipole is the most informative component of a hydrodynamic stimulus for source localisation with ALLs because the higher terms of a multipole expansion decay with distance more quickly [22]. The lower term—the monopole—is measurable at larger distances. However, it is driven by changes in an object’s size, so it is less relevant for localising moving objects. The dipole stimulus has become a popular source for studies with ALLs [5,6,7,8,9,13,14,23,24,25,26,27,28].

A potential flow model was used to simulate fluid flows produced by a sphere, usually referred to as a dipole field. This model was utilised in several other studies [9,13,25,28] and its usefulness is supported by experimental findings in fish lateral line research [5,25]. Potential flow velocity v is computed by [16]:(1)$$v=\frac{a^{3}}{{2||r||}^{3}}\left(-w+3r\frac{\left(w\times r\right)}{{\left||r|\right|}^{2}}\right),$$
where *a* is the radius of the sphere, $w=\langle w_{x},w_{y}\rangle$ are the instantaneous velocity components of the moving sphere in 2D, and r=s−p is the location of the sensor $s=\langle x,y\rangle$ relative to the source $p=\langle b,d\rangle$. The dipole’s position p(t) and velocity w(t) over time were computed as:(2)$$p\left(t\right)=p_{0}+A\left[\begin{array}{c} \cos\left(\varphi \right)\\ \sin\left(\varphi \right) \end{array}\right]\sin\left(2\pi ft\right),$$
and
(3)$$w\left(t\right)=\frac{dp(t)}{dt}=2\pi fA\left[\begin{array}{c} \cos\left(\varphi \right)\\ \sin\left(\varphi \right) \end{array}\right]\cos\left(2\pi ft\right),$$
where *A* is the amplitude and *f* the frequency of the oscillation, $p_{0}$ is the average position of the source, φ is the azimuth angle of the motion, and *t* indicates time.

We treat localisation as recovering the source’s average position $p_{0}$ from ALL velocity measurements over a period of time. The source’s movement during this period did not influence our results because there were an integer number of oscillations in each period. In other words, $p_{0}$ corresponded precisely with the average position in the measurement segments.

Even though a unique mapping exists between source states and their velocity profiles—i.e., the patterns an infinite continuous linear array of flow sensors would measure—the inverse problem is challenging because sensor arrays only capture a discrete segment of the velocity profile, which may not contain the informative zero-crossings or peaks. Figure 1 shows the parallel ($v_{x}$) and perpendicular ($v_{y}$) velocity profiles relative to the sensor array. These velocity profiles broaden and their amplitude decays with the distance of the source, reducing the information captured by a fixed-sized ALL. 2D sensitive sensors increase the chance of capturing one of the velocity profile’s more informative points because the zero-crossings of one velocity component are located near the peaks of the other velocity component.

Another challenge arises in the movement direction estimation. The velocity profiles of objects in the same place but moving in opposite directions differ only in their sign. Consequently, some of the dipole localisation algorithms struggle to differentiate between these source states. We employ a post-processing step to improve these algorithms’ movement direction estimation (see Section 2.5).

### 2.2. Simulation Environment

Eight sensors were simulated, based on the configuration of Wolf et al. [10]. The sensors were placed on the *x*-axis, centred around x=0. The length of the sensor array was L=40 cm, with 5.71 cm between sensors. The source sphere had a radius of a=10 mm and moved with an amplitude of A=2 mm at a fixed frequency of f=45 Hz. Similar frequency values have been used in the literature: 40 Hz in [7,8,23], 45 Hz in [13,27], and 50 Hz in [26,29]. The source’s radius and vibration amplitude are comparable to the work of Abdulsadda and Tan [7,8,24] (a=9.5 mm and A=1.91 mm). Figure 2 shows a schematic view of the present configuration. The source sphere was located between x=±0.5 m and from y=0.025 m to y=0.525 m, ensuring its edge was always at least 15 mm away from the closest sensor’s centre. The orientation of the source oscillation ranged from φ=0 rad to φ=2π rad. For each measurement, the fluid velocity at the sensors was simulated for a duration of T=1 s and sampled at 2048 Hz, comparable to the values used by Pandya et al. [6] (T=0.5 s at 2048 Hz). The simulated sensors had a Gaussian sampled velocity-equivalent noise of $\sigma =1.0\times 10^{-5}$ m/s. This value was chosen to be in the top five most sensitive sensors reported by Jiang et al. [30]: $2.5\times 10^{-6}$ m/s [31], $5\times 10^{-6}$ m/s at resonance [20], $8\times 10^{-6}$ m/s [32], $8.2\times 10^{-6}$ m/s [33] and $2\times 10^{-4}$ m/s [34]. The resulting signal to noise ratio (SNRs) are shown for the fifth sensor from the left in Figure 3.

### 2.3. Performance Analyses

Two methods of analysis were employed in this research to compare the performance of the dipole localisation algorithms. Analysis Method 1 varied the number of measurements the algorithms could use to train and optimise their hyperparameters. Not every algorithm requires training or has hyperparameters to optimise. For the algorithms that do not have a training phase, we expect a consistent performance regardless of the amount of data. The other algorithms are expected to improve as the amount of training data increases. In this analysis method, all sensors were sensitive to both the parallel and perpendicular velocity component. Analysis Method 2 varied which velocity components were measured by the sensors: (x + y) both components on all sensors, (x|y) alternating $v_{x}$ and $v_{y}$ for subsequent sensors, (x) only $v_{x}$ by all sensors, (y) only $v_{y}$ by all sensors. In this analysis method, the largest training and optimisation set was used.

The best performing algorithms based on the two analysis methods were also evaluated using higher velocity-equivalent noise levels to show how they perform on sensors with a lower SNR. The increased noise levels (σ) were: $1.0\times 10^{-3}$ m/s and $1.8\times 10^{-2}$ m/s based on [35,36], respectively, as cited in [30]. $1.0\times 10^{-4}$ m/s was added to bridge the gap with the analysis methods’ noise-level, which was $1.0\times 10^{-5}$ m/s. All sensors were sensitive to both the parallel and perpendicular velocity component, and the largest training set was used in this comparison.

In all analysis methods, the algorithms’ predictions were recorded for each measurement in the test set. A source state with a unique combination of position $p=\langle b,d\rangle$ and movement direction φ was used to generate each measurement. The error of the predicted position $E_{p}$ (m) and movement direction $E_{\varphi }$ (rad) were computed as:(4)$$E_{p}\left(\hat{P},P\right)=\sqrt{{\left(\hat{b}-b\right)}^{2}+{\left(\hat{d}-d\right)}^{2}},$$
and
(5)$$E_{\varphi }\left(\hat{P},P\right)=\left|atan2\left(\sin\left(\hat{\varphi }-\varphi \right),\cos\left(\hat{\varphi }-\varphi \right)\right)\right|,$$
where $P=\langle b,d,\varphi \rangle$ are the actual properties of a test state and $\hat{P}=\langle \hat{b},\hat{d},\hat{\varphi }\rangle$ is the prediction based on the test state’s velocity measurements. The areas in which the localisation algorithms’ median position errors were lower than 1 cm, 3 cm, 5 cm, and 9 cm were computed by discretising the simulated domain in 2×2 cm cells. The areas for the movement direction error were computed similarly with 0.01π rad, 0.03π rad, 0.05π rad, 0.09π rad.

The training and test sets contained randomly sampled source states. Poisson Disc sampling [37] was used to ensure an even spread of states over the simulated domain. The minimum distance between states $D_{s}$ controlled the number of source states in each set. This distance was computed as the Euclidean distance in the *x*–*y*–φ/2π space containing all possible source states. It can be interpreted as follows: when two states have the same position, their movement direction differs by at least $2\pi D_{s}$ rad; when they have the same movement direction, their position differs by at least $D_{s}$ m. The orientation dimension was divided by 2π to balance the number of positions and orientations considered. Table 1 shows the minimum sampling distance, the resulting number of states, and the average minimum distances in terms of position and orientation for each data set. The values of $D_{s}$ were chosen in terms of the source radius and correspond to the thresholds applied to the position and movement direction errors.

### 2.4. Parameter Optimisation Approach

Several of the dipole localisation algorithms have hyperparameters, which can be varied to fine-tune their performance. A single error metric that combines and balances both the position and movement direction is required to optimise these parameters. Given that we compare the dipole localisation algorithms based on the area in which they can accurately predict an object’s state, the hyperparameter optimisation process should prioritise perfecting accurate predictions over reducing the error of inaccurate predictions. Therefore, we used a normalised absolute error metric $E_{norm}$:(6)$$E_{norm}\left(\hat{P},P\right)=\frac{|\hat{b}-b|}{1\;m}+\frac{|\hat{d}-d|}{0.5\;m}+\frac{|E_{\varphi }\left(\hat{P},P\right)|}{2\pi \;\mathrm{rad}},$$
where $P=\langle b,d,\varphi \rangle$ are the properties of the test state and $\hat{P}=\langle \hat{b},\hat{d},\hat{\varphi }\rangle$ is the prediction. Compared to an error metric based on the Euclidean distance, the minimisation of $E_{norm}$ is less sensitive to predictions with a large error.

Algorithms that required training were optimised using 5-fold cross-validation (80% training/20% validation split). The other algorithms used the entire training set for a single validation pass. The hyperparameter values that minimised the mean validation error $E_{norm}$ were used in the evaluation with the withheld test set.

Appendix A provides the optimal values of all hyperparameters for each condition of the first two analysis methods. In Analysis Method 3, the algorithms used the optimal values of Analysis Method 1’s $D_{s}=0.01$ condition.

### 2.5. Dipole Localisation Algorithms

Each dipole localisation algorithm had access to a training set $T={\left\{\langle V^{i},P^{i}\rangle \right\}}_{i}^{N}$, where ${V^{i}}_{\left(sensors\times time\right)}$ are the velocity measurements over time and $P^{i}=\langle b^{i},d^{i},\varphi^{i}\rangle$ the state of the *i*th training dipole. The velocity measurements contain values for $v_{x}$ and $v_{y}$ depending on analysis variation. The to-be-predicted velocity profile is denoted as ${\tilde{V}}_{\left(sensors\times time\right)}$.

Not every algorithm explicitly used the time component of the velocity measurements. When required, the time dimension was averaged out by computing a DFT of the signal at each sensor. The magnitude of the 45 Hz components multiplied by the sign of their phase reconstructs the velocity signal over the sensors $v_{\left(sensors\times 1\right)}$. Abdulsadda and Tan [24] used this method—without multiplying the sign of the phase—to reduce the influence of noise. A Hamming window was used for computing the DFT.

In the following subsections, each dipole localisation algorithm is described. Table 2 provides a summary of their properties.

#### 2.5.1. The Random Predictor (RND)

The random predictor is used as a baseline for comparing performance. The algorithm does not use the training set T nor the to-be-predicted velocity measurement $\tilde{V}$. Instead, it generates a uniform random position and movement direction within the simulated domain (see Figure 2) for every test state.

#### 2.5.2. Linear Constraint Minimum Variance (LCMV) Beamforming

LCMV beamforming was introduced for dipole localisation by Yang et al. [12] and Nguyen et al. [13]. The algorithm computes a prediction by evaluating an energy value $E^{i}$ of each source state in the training set T [12,13]:(7)$$E^{i}=\frac{1}{{v^{i}}^{T}R^{-1}v^{i}},$$
where $R=\tilde{V}{\tilde{V}}^{T}$ is the covariance of the to-be-predicted source state, and $v^{i}$ is the *i*th training source’s velocity measurement with the time dimension averaged-out. The position and movement direction of the training source with the highest energy value is used as prediction $\hat{P}=\langle \hat{b},\hat{d},\hat{\varphi }\rangle$. LCMV is not able to differentiate between sources at the same position but with opposite orientations. To solve this issue, we computed the predicted state’s expected potential flow values for both the predicted and opposite movement direction. The movement direction with the smallest difference to the measured velocity was used as the final estimate.

#### 2.5.3. K-Nearest Neighbours (KNN)

The KNN algorithm generalises the template matching approach used by Pandya et al. [6]. KNN computes a prediction by finding the *k* training states with the most similar velocity measurements $V^{i}$ compared to the to-be-predicted source’s measurements $\tilde{V}$. Before computing the Euclidean distance between the velocity measurements, the time dimension was averaged out from both the training measurements and the measurement of the to-be-predicted source (see Section 2.5). Then, all velocity measurements were normalised by their maximum absolute value. Finally, the average position and movement direction of the *k* closest training states were computed and used as prediction $\hat{P}=\langle \hat{b},\hat{d},\hat{\varphi }\rangle$. The value of *k* was optimised, ranging from k=1 to k=20.

#### 2.5.4. The Continuous Wavelet Transform (CWT)

The CWT was introduced for dipole localisation by Ćurčić-Blake and van Netten [5]. The algorithm is based on the observation that potential flow and the pressure gradient along a lateral line can be expressed as wavelets. As in Wolf and van Netten [14], we extend the family of wavelets to include the perpendicular velocity component. Note, the coordinate system used here is slightly different. Deriving the wavelets from the potential flow formula (Equation (Equation 1))—using the approach of Ćurčić-Blake and van Netten [5]—finds (see Appendix B for the derivations):(8)$$\begin{array}{cc} v_{x} & =\frac{a^{3}\left||w|\right|}{{2|y-d|}^{3}}\left(\psi_{e}\cos\left(\varphi \right)+\psi_{o}\sin\left(\varphi \right)\right),\\ v_{y} & =\frac{a^{3}\left||w|\right|}{{2|y-d|}^{3}}\left(\psi_{o}\cos\left(\varphi \right)+\psi_{n}\sin\left(\varphi \right)\right), \end{array}$$
with
(9)$$\psi_{e}=\frac{2\rho^{2}-1}{{\left(\rho^{2}+1\right)}^{\left(5/2\right)}},$$
(10)$$\psi_{o}=\frac{3\rho }{{\left(\rho^{2}+1\right)}^{\left(5/2\right)}},$$
(11)$$\psi_{n}=\frac{2-\rho^{2}}{{\left(\rho^{2}+1\right)}^{\left(5/2\right)}},$$
(12)$$\rho =\frac{r_{x}}{r_{y}}=\frac{x-b}{y-d},$$
where a=1 cm is the radius of the source, $w=\langle w_{x},w_{y}\rangle$ is the movement velocity of the source, r=s−p is the relative location of the sensor $s=\langle x,y\rangle$ from the perspective of the source $p=\langle b,d\rangle$, and φ is the azimuth angle of the motion relative to the sensor array.

To compute a prediction, the CWT uses the to-be-localised velocity measurement with the time dimension averaged out $\tilde{v}$ (see Section 2.5) and the source positions $p^{i}=\langle b,d\rangle$ in the training set T. Note that the CWT does not use the training dipoles’ velocity measurements or movement directions. Instead, it computes the values of the wavelets for each position in the training set. These values were evaluated for an extended sensor array matching the simulation domain’s width and normalised by their maximum absolute value. Only the values at the eight sensors were kept after the normalisation step. From these values, a vector was constructed for each position $p^{i}=\langle b,d\rangle$ in the training set T, containing four CWT coefficients: one for each combination of velocity component and wavelet. The peak of a Gaussian surface fitted to this vector’s magnitude $Wv(p^{i})$ was used to estimate the sources’ position $\hat{p}=\langle b,d\rangle$. This fit was constrained to have a peak within the simulated domain (see Figure 2), and only Wv’s values between a factor $t_{min}$ and $t_{max}$ of its maximum were used for the fit. The values of $t_{min}$ and $t_{max}$ were optimised, ranging from 0 to 1 under the constraint that $t_{min}<t_{max}$. The movement direction was estimated as the circular mean of:(13)$$\begin{array}{cc} {\hat{\varphi }}_{x} & =-\mathrm{atan}c_{x}\frac{Wv_{x}^{o}\left(\hat{p}\right)}{Wv_{x}^{e}\left(\hat{p}\right)},\\ {\hat{\varphi }}_{y} & =-\mathrm{atan}c_{y}\frac{Wv_{y}^{n}\left(\hat{p}\right)}{Wv_{y}^{o}\left(\hat{p}\right)}, \end{array}$$
where $Wv_{n}^{m}\left(\hat{p}\right)$ is the CWT coefficient of ${\tilde{v}}_{n=(x\;\mathrm{or}\;y)}$ with $\psi_{m=(e,\;o,\;\mathrm{or}\;n)}$. Appendix C shows how these equations were derived and provides the analytical values of $c_{x}$ and $c_{y}$. Unfortunately, the values of $c_{x}$ and $c_{y}$ can be shown to depend on the source position for our simulated finite and discontinuous sensor array. Therefore, we optimised the values of $c_{x}$ (between 0.5 and 1) and $c_{y}$ (between 0.3 and 0.9). Consequently, the estimation of the movement direction was optimised for the positions where the CWT is accurate.

In total, four hyperparameters were optimised for the CWT ($t_{min}$, $t_{max}$, $c_{x}$, $c_{y}$). The best combination of values was determined in 30 iterations of the Bayesian optimisation algorithm provided by MATLAB [38].

#### 2.5.5. The Extreme Learning Machine (ELM)

The ELM is a neural network designed to provide “the best generalisation performance at extremely fast learning speed” [39]. An ELM is a single layer feed-forward network with randomly initialised weights on the hidden layer. These weights are not changed during training. To find the optimal weights for the output layer, the ELM can be treated as a linear system because the input weights and activation function are fixed. The online sequential ELM variant (OS-ELM) [40] was used to support iterative training. Random initial weights were generated, and the network was trained on the training set T. The velocity measurements were pre-processed by averaging out the time dimension and normalising with their maximum absolute value. Rectified linear units were used as hidden nodes, as recommended in Goodfellow et al. [41] (p. 168). The layer size $\bar{n}$ of the ELM was optimised using 101 values spaced logarithmically ranging from $\bar{n}=10$ to $\bar{n}=N$, where *N* is the number of training measurements in the training set T (see Table 1). The parameter sweep was terminated for the first value of $\bar{n}$ for which the ELM could not be trained due to a singular matrix inversion.

#### 2.5.6. The Multi-Layer Perceptron (MLP)

An MLP was implemented to determine how much better a high capacity network performs compared to the ELM. Rectified linear units were used as hidden nodes, as recommended in Goodfellow et al. [41] (p. 168). Linear activation functions were used on the input and output nodes. The weights of the network were initialised using normalised initialisation, introduced by Glorot and Bengio [42]. Bias-weights were initialised to zero and kept constant during training, essentially disabling them. The network was trained to minimise the mean absolute error (MAE) between the predicted source states ${\hat{P}}^{i}=\langle {\hat{b}}^{i},{\hat{d}}^{i},{\hat{\varphi }}^{i}\rangle$ and actual states $P^{i}=\langle b^{i},d^{i},\varphi^{i}\rangle$. This error metric differs from $E_{norm}$ (Equation (Equation 6)) because the MAE does not normalise the individual dimensions and does not consider the circular nature of φ.

As indicated earlier, 80% of the training set T was used for training, the remainder for validation. Each velocity measurement was pre-processed by averaging out the time dimension and normalising by its maximum absolute value. The Adam [43] optimisation algorithm was used with the recommended values for the gradient decay factor $\rho_{1}=0.9$, the squared gradient decay factor $\rho_{2}=0.999$, and denominator offset $\delta =10^{-8}$ [41] (p. 301). Weight decay was applied with a factor of $10^{-4}$. A decay factor of 0.1 was applied to the learning rate ϵ every 10 epochs. The remaining 20% of the training set was used to compute a validation error every 50 iterations. The validation set and training set were each shuffled every epoch. The training was stopped when the validation error did not reach a new minimum in the last 5 evaluations, or the number of epochs exceeded 500. The minibatch size was 2048. The network was pre-trained in three stages to improve the performance on source states close to the sensors, first using states within 20 cm, then 40 cm, and finally 60 cm of the origin.

Three hyperparameters were optimised: the learning rate ϵ ranging from $\epsilon =10^{-4}$ to $\epsilon =10^{-1}$, the number of layers *n* ranging from n=1 to n=4, and the layer sizes $\bar{n}$ ranging from $\bar{n}=16$ to $\bar{n}=1024$. Each layer had the same number of nodes to simplify the optimisation. The best combination of parameters was determined in 30 iterations of the Bayesian optimisation algorithm provided by MATLAB [38].

#### 2.5.7. The Gauss–Newton (GN) Algorithm

GN was implemented for dipole localisation by Abdulsadda and Tan [8]. The algorithm does not use the training set T. Instead, it iteratively fits a potential flow model to the absolute value of the to-be-predicted source state’s velocity measurements with the time dimension averaged out $|\hat{v}|$. Let $\theta_{0}=\langle b_{0},d_{0},\varphi_{0}\rangle$ be an initial estimate. Then the next iteration is given by [8]: (14)$$\theta_{k+1}=\theta_{k}+\lambda {\left({\nabla |v(\theta_{k})|}^{T}\nabla \left|v\left(\theta_{k}\right)\right|\right)}^{-1}{\nabla |v(\theta_{k})|}^{T}\left(\right|\hat{v}\left|-\left|v(\theta_{k})\right|\right),$$
where λ is a step size parameter, and $v(\theta_{k})$ is the potential flow of a source state $\theta_{k}$ computed using Equation (Equation 1). The algorithm terminates when the change in θ is smaller than $\epsilon =1\times 10^{-3}$, the number of iterations exceeds 100, or the matrix inversion could not be computed due to a singular matrix. The gradient of $\left|v(\theta_{k})\right|$ was estimated numerically using a step size of $\delta =1\times 10^{-3}$. The step size was λ=1, because every iteration solves a linearised version of the fitting problem (see [8]).

The simulated domain’s centre was used as the initial estimate ($b_{0}=0$ m and $\varphi_{0}=\pi$ rad). However, the centre of the *d*-domain may not be the optimal value for $d_{0}$ because states close to the sensors are typically localised more accurately, and the distance between the actual source position and the initial estimate influences the convergence of GN [8]. Therefore, the value of $d_{0}$ was optimised and chosen from the range $d_{0}=0.025$ m to $d_{0}=0.525$ m, with a step size 0.025 m. Differently from Abdulsadda and Tan [8], we applied an upper and lower bound on $\theta_{k}$. After every iteration, the values of $b_{k}$ and $d_{k}$ were clipped to the simulated domain (see Figure 2), and a modulo 2π operation was applied to $\varphi_{k}$.

A single post-processing step was performed to improve the movement direction estimation. GN is not able to differentiate between sources at the same position with opposite orientations because it uses the absolute velocity measurements to fit a potential flow. To solve this issue—as with the LCMV beamforming algorithm—we computed the predicted state’s expected potential flow values for both the predicted and opposite movement direction. The movement direction with the smallest difference to the measured velocity was used as the final estimate.

#### 2.5.8. The Newton–Raphson (NR) Algorithm

NR was also introduced for dipole localisation by Abdulsadda and Tan [8]. The algorithm is very similar to GN; only the hyperparameter optimisation and update step are different. Let $\theta_{0}=\langle b_{0},d_{0},\varphi_{0}\rangle$ be an initial estimate. Then, the next iteration is given by [8]:(15)$$\theta_{k+1}=\theta_{k}-\lambda {G(\theta_{k})}^{-1}g\left(\theta_{k}\right),$$
with
(16)$$g\left(\theta \right)={\nabla |v(\theta )|}^{T}\left(\right|\hat{v}|-|v\left(\theta \right)|),$$
and
(17)$$G\left(\theta \right)=\frac{\partial g(\theta )}{\partial \theta },$$
where λ is a step size parameter, $v(\theta_{k})$ is the potential flow of a source $\theta_{k}$ computed using Equation (Equation 1), and $|\hat{v}|$ is the absolute value of the to-be-predicted source state’s velocity measurements with the time dimension averaged out. The gradient g(θ) and Hessian G(θ) were estimated numerically with a step size of $\delta =1\times 10^{-3}$. The step size λ, stopping conditions, bounds check, and initial estimate were the same as for GN. Different from GN, a norm limit *l* was applied to the change in θ in each iteration. The value of *l* was optimised, ranging from l=0.1 to l=1. The best combination of $d_{0}$ and *l* was determined in 30 iterations of the Bayesian optimisation algorithm provided by MATLAB [38]. The same post-processing step was applied as in GN, to improve the movement direction estimation.

#### 2.5.9. The Least Square Curve Fit (LSQ) Algorithm

The LSQ predictor was implemented as another baseline for comparing performance. The algorithm does not require training and does not have hyperparameters to optimise. As a result, the training set T is not used by LSQ. Therefore, the number of training states does not influence the performance of LSQ. Consequently, the algorithm is only used in the second analysis method.

LSQ computes a prediction using the *lsqcurvefit* function from MATLAB [38]. The algorithm fits a potential flow model v(θ) (Equation (Equation 1)) to $\hat{v}$: the to-be-predicted source state’s velocity measurement with the time dimension averaged out. The *lsqcurvefit* function implements the *trust-region-reflective* algorithm (see [44]). Lower and upper bounds were provided for all elements of $\theta =\langle b,d,\varphi \rangle$, to limit their values to the simulated domain (see Section 2.2). Gradients were estimated numerically by *lsqcurvefit*, using a step size of $\delta =1\times 10^{-3}$. The algorithm terminated when the number of iterations exceeded 100 or the change in θ was smaller than $\epsilon =1\times 10^{-3}$. The function and optimality tolerance checks were set to machine-precision. The initial estimate $\theta_{0}$ is identical to GN.

#### 2.5.10. The Quadrature Method (QM) Algorithm

QM is a newly proposed localisation algorithm that takes advantage of 2D sensitive sensors. The algorithm combines measurements of $v_{x}$ and $v_{y}$ to construct a curve, $\psi_{quad}$, that has characteristics that are nearly independent of the orientation of a source:(18)$$\psi_{quad}=\sqrt{v_{x}^{2}+\frac{1}{2}v_{y}^{2}}.$$

Combining $v_{x}$ and $v_{y}$ in this way is somewhat analogous to computing the overall amplitude of two quadrature time signals. Appendix D explains the properties of the quadrature curve in more detail. Summarising, the factor 1/2 can be shown to negate the influence of the orientation φ at the maximum of the curve, allowing for an estimate of the lateral position *b* that is virtually independent of the orientation φ. The distance of the source *d* is linearly related to a measure of the width of $\psi_{quad}$. Two so-called anchor points $\rho_{anch}^{\pm }$ are used to measure this width. These anchor points are located where $\psi_{quad}$ takes a value of about 0.458 times the curve’s maximum and their location is almost independent of the dipole’s orientation. Note, ρ (Equation (Equation 12)) describes a source state’s location along the sensor array normalised by its distance. Practically, though, the locations of these anchor points are estimated in terms of *x*. Given the location of the anchor points, the source distance *d* is computed as:(19)$$d=\frac{1}{1.79}\left(\rho_{anch}^{+}-\rho_{anch}^{-}\right).$$

In practice, the locations of the anchor points were estimated by linearly interpolating $\psi_{quad}$ between the two sensors where $\psi_{quad}$ intersected with 0.458 times its maximum value.

An accurate estimate of the movement direction φ can be computed directly from the measured velocity values once the source state’s position $p=\langle b,d\rangle$ is known. For this purpose, the measured velocity is analysed using the wavelets from the CWT (see Section 2.5.4). Figure 4 visualises the wavelets and their form in a 3D $\psi_{e}$–$\psi_{o}$–ρ space. The movement direction can be recovered from $\psi_{e}$–$\psi_{o}$ slices of this 3D space at the sensor locations (Figure 4c). In these slices, the wavelets and the measured velocity are vectors (${\vec{\psi }}_{e}$, ${\vec{\psi }}_{o}$, and ${\vec{v}}_{x}$) with lengths corresponding to their values at the sensor. Note that all these values are known because the wavelets can be computed from the estimated position. The angle between ${\vec{v}}_{x}$ and ${\vec{\psi }}_{e}$ corresponds to the to-be-predicted movement direction φ. To compute φ, we use the vector combination of the wavelets ${\vec{\psi }}_{env}={\vec{\psi }}_{e}+{\vec{\psi }}_{o}$, which has a length $\psi_{env}=\sqrt{\psi_{e}^{2}+\psi_{o}^{2}}$. The angle between ${\vec{\psi }}_{env}$ and ${\vec{\psi }}_{e}$ is $\varphi^{\prime }=\mathrm{atan}\psi_{o}/\psi_{e}$. The difference between φ and $\varphi^{\prime }$ is $\alpha =\mathrm{acos}v_{x}/\psi_{env}$ because $v_{x}$ is a linear combination of the two wavelets. Consequently, the movement φ can be estimated using:(20)$$\varphi =\varphi^{\prime }\pm \alpha ,$$

Depending on the source’s position and movement direction and the sensors’ location, α should be added or subtracted from $\varphi^{\prime }$. Therefore, two estimates were computed for each sensor’s $v_{x}$ measurement. The circular median of all sixteen estimates was used as final prediction. This estimation approach also works for $v_{y}$ when using $\psi_{o}$ and $\psi_{n}$ in place of $\psi_{e}$ and $\psi_{o}$.

The predictions based on the anchor points’ estimated locations are limited because the spatial resolution of the sensor array limits the accuracy of the anchor point location estimation. In addition, one or both anchor points may not be within the measurable range of the ALL.

A refinement step was introduced to improve the predictions. First, the estimate of the position was improved by fitting a potential flow model to $\psi_{quad}$. The fit was computed using the *lsqcurvefit* function from MATLAB [38]. The position and orientation estimates computed using the anchor points were used as starting estimate $\theta_{0}=\langle b_{0},d_{0},\varphi_{0}\rangle$. Lower and upper bounds were provided for all elements of θ. Gradients were estimated numerically by *lsqcurvefit*, with a step size of $\delta =1\times 10^{-3}$. The fit procedure terminated when the number of iterations exceeded 10, or the change in θ was smaller than $\epsilon =1\times 10^{-3}$. The function and optimality tolerance checks were set to machine precision. Then, the orientation was re-estimated using the improved position estimate. The complete refinement step was repeated four times, each iteration using the previous estimated position and orientation as the starting point.

## 3. Results

In this research, ten dipole localisation algorithms were compared by the area in which they accurately estimate the position and movement direction of an object. Specifically, the area with a median position error $E_{p}$ below 1 cm, 3 cm, 5 cm, and 9 cm, and the area with a median orientation error $E_{\varphi }$ below 0.01π rad, 0.03π rad, 0.05π rad, and 0.09π rad are reported. Section 3.1 presents the first analysis’ results, where we varied the available data set size, and Section 3.2 presents the second analysis’ results, where the sensor sensitivity direction was varied. Section 3.3 provides additional results to compare the localisation algorithms, including the performance of the best three algorithms on simulated sensors with lower SNRs.

### 3.1. Analysis Method 1: Amount of Training and Optimisation Data

This analysis method varied the number of measurements the algorithms could use to train and optimise their hyperparameters to show how that influences the algorithms’ performance. LSQ was not included in this analysis because it does not require training nor has hyperparameters to optimise. However, LSQ’s results from the comparable (x + y) condition of Analysis Method 2 are shown to allow for a comparison of the performance of all algorithms.

The results of this analysis are summarised in Figure 5 and Figure 6. These figures visualise the total area in which the median position error and median orientation error were below their respective thresholds. There are several observations of note. Firstly, as expected, the model-based algorithms’ areas did not increase with the amount of training and optimisation data. With $D_{s}=0.09$, we found 0.21 m${}^{2}$ for QM, 0.22 m${}^{2}$ for GN, and 0.11 m${}^{2}$ for NR at $E_{p}\leq 1$ cm. In contrast, the template-based algorithms’ and the neural networks’ areas did increase with the training and optimisation sets. Using the largest data set $D_{s}=0.01$, these algorithms achieved a position error lower than 1 cm in areas of 0.18 m${}^{2}$ for MLP, 0.14 m${}^{2}$ for KNN, 0.12 m${}^{2}$ for LCMV, 0.1 m${}^{2}$ for ELM, and 0.00 m${}^{2}$ for CWT. Only the random predictor had a median position error larger than 9 cm everywhere.

Secondly, only QM, GN, and NR had a median orientation error below 0.01π rad in some part of the simulated domain with $D_{s}=0.09$, with areas of 0.06 m${}^{2}$, 0.06 m${}^{2}$, and 0.02 m${}^{2}$, respectively. With the largest training and optimisation set ($D_{s}=0.01$), the MLP and KNN reached the 0.03π rad threshold in 0.09 m${}^{2}$ and 0.02 m${}^{2}$, respectively, whereas LCMV only reached 0.05π rad in an area of 0.04 m${}^{2}$. The other predictors had a median orientation error larger than 0.09π rad everywhere.

The predictors’ performance is not only characterised by the area in which they work well; it is also important to show how accurate predictions were in areas where they did not work well. To provide a complete picture of the predictor performance, Figure 7 and Figure 8 visualise the distributions of the position and orientation errors, respectively.

Firstly, note that all dipole localisation algorithms had lower median position errors than the random predictor with all training and optimisation sets. This difference was smaller for the median orientation error. In particular, ELM’s orientation error distribution with the smallest training and optimisation set was similar to that of the random predictor. Secondly, the position error distribution of NR had a large tail; the 75th and 95th percentiles were 0.81 m and 0.98 m with $D_{s}=0.09$, respectively. The GN predictor also had a higher 95th percentile (0.32 m) than the otherwise similarly performing QM predictor (0.14 m) with $D_{s}=0.09$. The same relation was also present in the orientation error distributions. The 95th percentile of GN (2.28 rad) was larger than that of QM (0.92 rad) with $D_{s}=0.09$.

Another interesting observation is the difference between the development of the orientation error distributions of MLP and KNN. The errors of KNN decreased gradually with each increase in training and optimisation data, whereas the errors of the MLP decreased drastically between $D_{s}=0.03$ and $D_{s}=0.01$. Note that the MLP used four layers with $D_{s}=0.01$ and one layer with all the other training and optimisation sets (see Table A1). Finally—unlike the position error—the CWT’s orientation errors were larger with the largest optimisation set compared to the smaller sets.

### 3.2. Analysis Method 2: Sensor Sensitivity Axes

This analysis method varied which velocity components were measured by the sensors to determine how that influences the algorithms’ performance. The QM predictor was not included in this analysis because it requires both velocity components to be measured by all sensors. However, QM’s results from the comparable $D_{s}=0.01$ condition of Analysis Method 1 are shown to allow for a comparison of the performance of all algorithms.

The results of this analysis are summarised in Figure 9 and Figure 10. These figures visualise the total area in which the median position error and median movement direction error were below their respective thresholds. In configuration (x + y)—which measured both velocity components at all sensors—the GN predictor had the largest area with median position error lower or equal to 1 cm (0.22 m${}^{2}$). The MLP and LSQ predictors followed with 0.19 m${}^{2}$ and 0.18 m${}^{2}$, respectively. The NR predictor (0.11 m${}^{2}$) performed worse than KNN (0.14 m${}^{2}$) and LCMV (0.12 m${}^{2}$). The ELM did not perform well at the 1 cm threshold (0.01 m${}^{2}$). However, the areas at the higher thresholds were similar to those of the MLP. The CWT and RND predictors had a median position error larger than 1 cm in the entire simulated domain.

In the other configurations, LSQ performed the best; the median position error was lower or equal to 1 cm in 0.12 m${}^{2}$ in configuration (x), and 0.15 m${}^{2}$ in configuration (y) and (x|y). In general, the areas with median position errors lower or equal to 3 cm were larger with the alternating configuration (x|y) compared to configurations (x) or (y). The CWT predictor is an exception; it performed the best in configuration (x) at all position error thresholds. For GN and NR, the larger area at the 3 cm threshold in configuration (x|y) went along with a smaller area at the 1 cm threshold compared to configuration (x).

The movement direction errors, shown in Figure 10, indicate a slightly different pattern. The LSQ predictor had the largest areas with a median movement direction error below 0.01π rad in all configurations (0.05 m${}^{2}$ in (x) and (x|y), 0.07 m${}^{2}$ in (y), and 0.08 m${}^{2}$ in (x + y)). The GN predictor performed better than LSQ at the higher thresholds in configuration (x + y) (0.19 m${}^{2}$ to 0.15 m${}^{2}$ at $E_{\varphi }\leq 0.05\pi$ rad and 0.26 m${}^{2}$ to 0.18 m${}^{2}$ at $E_{\varphi }\leq 0.09\pi$ rad). Interestingly, the MLP performed better than KNN, especially in configuration (x|y) and (x + y) (see Figure 10).

Similar to the first analysis (Section 3.1), we also show the error distributions in Figure 11 and Figure 12. The benefit of using both $v_{x}$ and $v_{y}$ (x + y) is visible in the lower median, 75th, and 95th percentiles. The alternating configuration (x|y) also resulted in lower median, 75th, and 95th percentile values compared to configurations (x) and (y), except for the NR, CWT and RND predictors. Similar to the results in Analysis Method 1, the MLP, KNN, and ELM predictors have lower 95th percentiles in the position error distribution in configuration (x + y) (0.13 m, 0.17 m, and 0.16 m, respectively) than the model-based algorithms (0.32 m for GN, 0.82 m for LSQ, 0.98 m for NR).

### 3.3. Additional Results

This section presents additional results of the two analysis methods and the performance of the three best algorithms on simulated sensors with lower SNRs. The algorithms’ training and prediction times are reported, and spatial and polar maps of the median errors are shown, visualising the areas in which the predictors performed well and the effect of the source’s orientation on the prediction accuracy.

Table 3 shows each localisation algorithms’ average prediction time using both velocity components and total training time for the largest training set ($D_{s}=0.01$). All algorithms were evaluated on a high performance computing cluster, using 12 cores of an Intel Xeon 2.6 GHz processor and 64 GB RAM. The run-time performance of the implementations was optimised for computing many predictions simultaneously. As a result, the average prediction times may not be representative of the single-source prediction time. The random predictor had the shortest prediction time, followed by the MLP and ELM. KNN was also one of the quicker algorithms, about three times slower than the MLP. The model-based predictors were slower: the MLP could compute roughly four, eight and nine predictions in the time it took for GN, LSQ and QM to compute a prediction. The remaining algorithms were considerably slower than the MLP (36 times for LCMV, 175 times for NR, and 300 times for CWT). Most of the CWT’s prediction time came from fitting the Gaussian to the coefficients. It should be noted that the computational aspects of our implementations have not been extensively optimised, and the degree of optimisation between algorithms may have varied.

Figure 13 shows the spatial contours of the median position error $E_{p}$ and median movement direction error $E_{\varphi }$. These figures show in which areas the algorithms were able to compute an accurate prediction. Both QM and GN had a median position error within 1 cm up to roughly 35 cm from the sensor array. The GN predictor was better at locating source states in the lower corners of the simulated domain than LSQ and QM. However, its predictions directly in front of the sensors were worse. The MLP and KNN also performed well in the lower corners of the simulated domain. However, their median position error was lower than 1 cm up to only roughly 25 cm. For the LSQ and LCMV predictors, the median position errors were smaller than 1 cm up to 30 cm and 20 cm, respectively. These algorithms showed a quick drop in performance as the distance of a source state increases. Their median position errors were larger than 9 cm from 30 cm and 45 cm of the sensor array, respectively. QM, GN and the MLP had a median position error lower than 9 cm at least up to 50 cm. The contours in the orientation errors were similar. Comparing the two neural networks, the MLP had more accurate movement direction predictions than the ELM and also covered a larger area of the simulated domain.

Polar contours of the median position and median movement direction errors are shown in Figure 14, indicating the effect of source movement direction φ and distance *d* on the predictions. QM and GN performed similarly in the (x + y) configuration; both algorithms had accurate position and movement directions predictions for all source orientations φ. In configuration (x + y), QM’s movement direction prediction was slightly more accurate at longer distances for parallel and perpendicular movement directions. Closer to the sensors, QM’s movement direction prediction was more accurate for source states with a parallel orientation. GN had accurate movement direction predictions for parallel source states at all distances in configuration (x + y). The difference in movement direction estimation between QM and GN shown in Figure 14 can be explained by GN’s wider range of accurate predictions visible in Figure 13. LSQ had trouble predicting the position and movement direction of source states with orientations between $\varphi =\pm \frac{1}{4}\pi$ rad—except for parallel sources φ=0 rad—in configurations (x + y) and (y). GN’s localisation performance in configurations (x) and (y) is interesting as well, as it favoured perpendicular source states. On the other hand, GN’s movement direction predictions in configuration (y) were inaccurate regardless of the source state’s orientation φ and distance *d*.

Figure 15 shows how QM, GN, and the MLP perform using sensors with lower SNRs. Only the position error is shown. Appendix E contains a similar figure for the movement direction errors, as well as the spatial and polar projections. QM has the largest area with median position errors below 1 cm at the higher noise levels. However, from $\sigma =1\times 10^{-4}$ m/s and up, the MLP has lower median and 75 percentile values.

## 4. Discussion

In Analysis Method 1, we confirmed that (1) the total area in which the model-based algorithms produce accurate predictions does not depend on the amount of training and optimisation data, and (2) the template-based algorithms’ and neural networks’ areas with accurate predictions increase with the amount of training and optimisation data (Figure 5 and Figure 6). The MLP, in particular, benefits from large amounts of training data. However, only with our largest training and optimisation set (90,435 states) are the MLP and KNN able to approach QM’s and GN’s performance. Even in that case, their movement direction predictions are no match for those of QM and GN. Whether the MLP or KNN provides the better performance depends on the amount of training and optimisation data. The MLP performed better with our smallest and largest sets (Section 3.1). In the other cases, KNN performed better.

In Analysis Method 2, we demonstrated the benefit of 2D sensitive sensors compared to 1D sensitive sensors. Improved position and movement direction estimation of ELMs using 2D sensors compared to using only $v_{x}$ was previously shown by Wolf and van Netten [45]. In the present study, we showed that other localisation algorithms also benefit from using both velocity components. In addition, we demonstrated that the alternating sensor configuration (x|y) used by Yang et al. [12] and Nguyen et al. [13] is not an adequate substitute for 2D sensitive sensors at the used spatial resolution. However, it does improve performance compared to measuring a single velocity component along the entire sensor array. We speculate that the performance difference between the alternating sensor configuration and 2D sensitive sensors diminishes as the sensor array’s spatial resolution increases.

Finally, we showed that the newly introduced QM provided the best overall performance with 2D sensors, regardless of the data set size. Its areas with a median position error below 1 cm and median movement direction error below 0.01π rad were as large as that of GN (Figure 5 and Figure 6). However, the tails of the error distributions of QM were shorter than those of GN (Figure 7 and Figure 8). In other words, the less-accurate predictions of QM were better than those of GN. With 1D sensors, LSQ performed the best, except for source states with an orientation close to φ=0 rad.

It should be noted that the model-based algorithms depend on an accurate forward model. Fitting a potential flow model only works when it accurately describes the actual velocity measurements. When more complicated hydrodynamic phenomena are present in the measurements, a forward model may become quite complex. In that case, the MLP and KNN may be good alternatives. However, these algorithms require a lot of training and optimisation data to reach a similar performance as the model-based algorithms.

We continue the discussion with remarks specific to the used algorithms in the following subsections.

### 4.1. The Gauss–Newton (GN) Algorithm

GN’s performance in this work was mostly in line with the results presented by Abdulsadda and Tan [8]. They showed that GN has a superior localisation performance compared to LCMV beamforming and template matching. In addition, they highlighted that the convergence behaviour of GN heavily depends on the initial estimate. In their simulation, the largest region of convergence had a radius of 1.5 cm, depending on the source’s position [8]. In the current study, we did not estimate the region of convergence. Instead, we showed that—based on the median position and movement direction errors in the simulated domain—GN can be used in a larger area, especially when 2D sensitive sensors are used. However, there is no guarantee of convergence in that larger area, as is evident from the large tail in the error distributions (Figure 11 and Figure 12). To improve the convergence rate, one could potentially use the estimated position and movement direction of another prediction method as the initial estimate of GN.

A difference in our results compared to Abdulsadda and Tan [8], is the effect of the source state’s orientation on the prediction accuracy when only $v_{x}$ or only $v_{y}$ are used (Figure 14). We found that parallel source states are localised less accurately than other orientations when only $v_{x}$ is used. Abdulsadda and Tan [8] did not report such an effect. In their results, of the 19 states reported, four had a roughly parallel orientation. Due to those state’s positions it is difficult to determine how much their movement direction influenced the prediction accuracy. So, this effect may not have been observable with their experimental setup.

Another difference compared to Abdulsadda and Tan [8] were our step tolerance and maximum number of iteration hyperparameters. Both parameters determine the accuracy and computational cost of the algorithm. For our experiments, a prediction within 1 mm and 0.001π rad was sufficient. The maximum number of iterations was reduced to 100 compared to the 2500 used by Abdulsadda and Tan [8]. We found that the limit of 100 iterations did not prevent a prediction from reaching the outer edges of the simulated domain. Depending on the use case, these parameters can be tuned to provide the best performance.

Finally, unlike Abdulsadda and Tan [8], we applied a bounds check on each iteration’s estimated position and movement direction, to keep the estimates within the simulated domain (Figure 2). Since we did not compare GN’s performance with and without the bounds check, we cannot draw conclusions about its effect. However, we expect that the bounds check has two advantageous effects. Firstly, it may cause non-converging predictions to terminate earlier because—depending on their trajectory—the difference between estimates in subsequent iterations may become smaller than $\epsilon =1.0\times 10^{-3}$. Secondly, it may cause some otherwise non-converging predictions to converge. Instead of continuing on their trajectory, these estimates move along the simulated domains boundaries and find a path towards the solution.

### 4.2. The Newton–Raphson (NR) Algorithm

Unlike the results of Abdulsadda and Tan [8], NR did not perform similar to GN in our analyses. Computing NR’s predictions took more than 40 times as long as for GN (Table 3). The increased run-time cost of NR in our implementation is probably due to the numerical estimation of the Hessian. The difference in localisation performance may also be due this numerical estimation. The Newton method (Equation (Equation 15)) requires that the Hessian is positive definite; otherwise, it may move the prediction towards a saddle point instead of a minimum [41] (p. 279,304). Manual inspection of the Hessian found negative eigenvalues in all iterations for the inspected source states, regardless of the prediction accuracy. A preliminary hyperparameter validation showed that the regularisation strategy suggested by Goodfellow et al. [41] (p. 304) to solve this issue did not improve the performance of NR. From their publication, it is not clear whether Abdulsadda and Tan [8] computed the Hessian analytically. Alternatively, this difference in performance may be explained if their initial estimates were within NR’s region of convergence.

In our implementation of NR, we applied the same bounds check as for GN. Since we did not compare the performance of NR with and without the bounds check, we cannot draw conclusions about its effect. However, we expect the same improvements as with GN.

### 4.3. The Multi-Layer Perceptron (MLP) and Extreme Learning Machine (ELM)

We showed that an MLP performs quite a lot better than an ELM when a large amount of training data is available (90,435 source states) (Figure 9 and Figure 10). The main differences between these neural networks are the training procedure and the higher capacity of the MLP. The MLP was trained to minimise the mean absolute error (MAE), whereas the ELM minimises the mean squared error (MSE). The difference in capacity was especially large when the largest training and optimisation set was used with 2D sensitive sensors. In that case, the MLP had roughly ten times more weights than the ELM and roughly sixty times more trainable weights. When the smaller training and optimisation sets were used, the MLP performed more similar to the ELM and used only a single layer.

The comparison of QM, GN and the MLP on simulated sensors with lower SNRs showed the robustness of the MLP against noise (Figure 15). This finding is in line with Boulogne et al. [9], who showed that the MLP was more robust to noise than ELMs and echo state network (ESNs).

The performance of the MLP may be improved further by also training bias-weights. Bias-weights serve as an activation-offset for each node in a layer, providing additional flexibility in the activation function. Bias-weights increase the capacity of the network and enable nodes to output non-zero values when their input is zero. We expect that the network’s prediction of, in particular, the source distance benefits from bias-weights because the distances are not centred around the same value as the input.

### 4.4. The Quadrature Method (QM) Algorithm

In this study, we introduced the QM dipole localisation algorithm. The algorithm is designed for 2D sensitive sensors and provides state-of-the-art performance. Compared to GN, the algorithm produces more accurate predictions close to the sensor array (Figure 13) and has less skewed error distributions (Figure 7 and Figure 8). These results indicate that the iterative refinement procedure of QM converges better than the non-linear optimisation of GN. In addition, the effective area of QM was larger than that of GN when using simulated sensors with lower SNRs (Figure 15).

Aside from its performance, QM has several attractive attributes. For instance, the orientation estimation procedure is very quick—with a computational complexity in the order of sensors—and only depends on the measured velocity and an estimate of the source’s position. As a result, any algorithm that estimates a source’s position can use QM’s orientation estimation. In addition, QM is able to compute an initial position estimate directly from the measured velocity (also with a computational complexity in the order of sensors). Consequently, QM does not require a hyperparameter specifying an arbitrary initial estimate. It should be noted that this initial estimate is accurate only in a limited area, as it depends on anchor-points on $\psi_{quad}$ that have to fall within the sensor array (Appendix D). Finally, the run-time of QM is easily tuned using two simple hyperparameters: the number of refinement iterations and the number of iterations used to fit the potential flow model. The hyperparameters values used in this study resulted in an average prediction time that was roughly two and a half times slower than GN.

Several interesting research questions about QM remain unanswered. For instance, we did not analyse the QM’s performance using only its initial estimate. Especially for higher resolution sensor arrays, these estimates may perform quite well compared to the other algorithms. In addition, other algorithms may benefit from using QM’s initial estimate. For example, GN’s performance close to the sensor array may improve when QM’s initial estimate is used as the starting point for fitting the potential flow model. Other two-stage combinations of algorithms may also be interesting to develop.

### 4.5. Future Research Directions and Possible Applications

In the present research, ten dipole localisation algorithms were compared using a stationary ALL in a 2D environment. Applications of ALLs typically operate in more complex environments that are embedded in 3D and may include self-motion. The shape and movement of a sensing platform itself would have a significant impact on the flow fields which were omitted in this analysis (see, for instance, Windsor et al. [46] for an analysis of flow fields around gliding blind cave fish). Localising sources in 3D requires different sensor configurations. Yang et al. [12] used a sensor array consisting of two orthogonal lines on a cylinder to demonstrate LCMV beamforming’s 3D localisation performance. Wolf et al. [15] used two parallel ALLs to localise multiple simultaneous sources in 3D. Analysing QM’s performance on 3D localisation tasks would be an interesting future research project.

Other interesting future research directions include using more realistic fluid flow simulations. For instance, Lin et al. [47] used a Navier–Stokes equation solver to predict the ratio of a source’s distance and size based on the measured velocity amplitude range. Additionally, it remains unclear how well the algorithms’ performances transfer to localising differently shaped or self-propelled objects. Finally, the algorithms could be compared on their performance locating moving objects instead of stationary dipoles.

## 5. Conclusions

In the present study, we compared a wide range of algorithms for determining a dipole’s position and orientation in the vicinity of a flow sensor array. To demonstrate the effect of the amount of available data to optimise or tune each algorithm, we sampled a bounded domain with four levels of granularity. To demonstrate the effects of sensitivity directions of the flow sensors, we extended the implementation of existing algorithms to support data from four different sensor configurations: sensitivity parallel to the array, sensitivity at a right angle to the array, sensors with alternating sensitivity directions, and finally an array of 2D-sensitive sensors. These effects on the algorithms were quantified by the area in which an algorithm can correctly determine a dipole’s position and orientation relative to the array with predefined degrees of accuracy. For further comparisons, we also disclosed box plots to indicate the distribution of errors, as well as visual representations of these errors in the 2D spatial domain and source orientation domain.

We demonstrated the benefit of 2D-sensitive sensors compared to 1D sensitive sensors for a dipole localisation task. All considered algorithms benefited from using information from 2D-sensitive sensors, although the amount of improvement varies. The extension to 2D-sensitive sensors allowed the introduction of a novel dipole localisation algorithm coined the quadrature method (QM). This algorithm is designed to take advantage of geometric properties that result from 2D-sensitive flow measurements. We showed that QM provides state-of-the-art performance and produces more accurate predictions than the Gauss–Newton (GN) algorithm [8], especially for source positions close to the sensor array.

Finally, we analysed how dipole localisation algorithms’ performance depends on the amount of training and optimisation data. We find that template-based algorithms and neural-networks require large amounts of training data to approach the performance of model-based algorithms that require only a small training and optimisation set in a simulation setting.

Since the simulation’s assumptions are all based on the potential flow of a vibrating sphere, the resulting flow fields can be used for all dipole fields. Such fields are not restricted to those generated by submerged moving objects. Exactly the same dipole field equations are associated with acoustic, electric, and magnetic phenomena. Therefore, the comparisons made in the present work may also be of interest in other applications than hydrodynamic imaging.

## Figures and Tables

**Figure 1 sensors-21-04558-f001:**
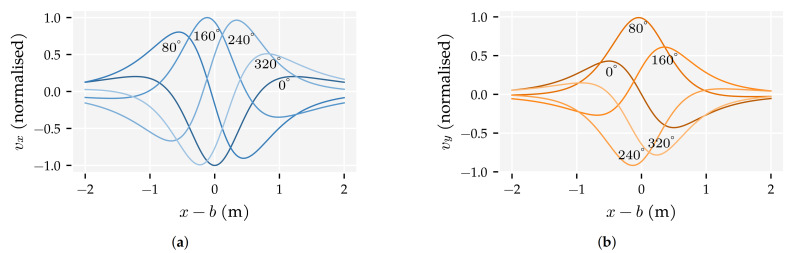
Normalised continuous velocity profiles for five movement directions (indicated) of a source at *d* = 1 m from the sensor array’s centre: (**a**) *v_x_* and (**b**) *v_y_*. The sensors are located along the *x* axis and *b* is the source sphere is *x* position in m.

**Figure 2 sensors-21-04558-f002:**
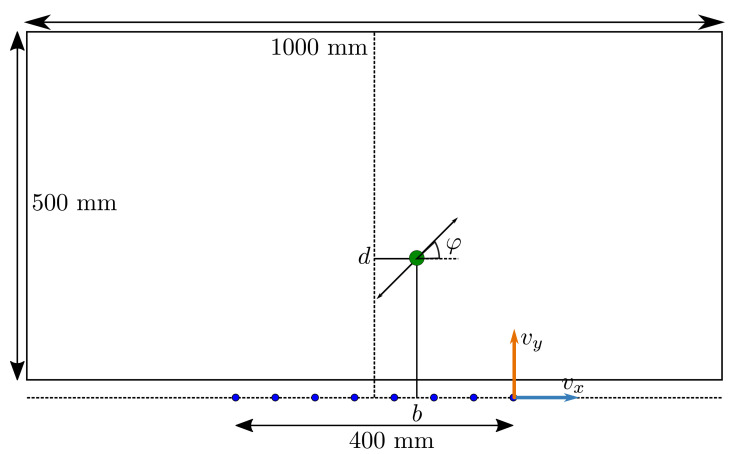
A schematic view of the simulated environment. The source sphere (green) has a radius of 1 cm and is shown to scale. A possible movement direction is shown by the arrow (not in scale). The sensor locations are shown in blue. Parallel $v_{x}$ and perpendicular $v_{y}$ velocity components are indicated at the right-most sensor (not to scale). The area in which the source sphere is positioned is offset by 25 mm from the array location, ensuring a minimal distance of 15 mm between the source’s edge and closest sensor’s centre.

**Figure 3 sensors-21-04558-f003:**
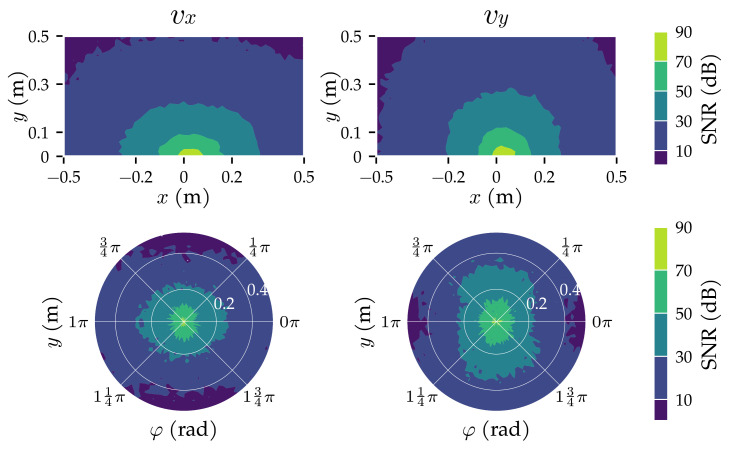
The signal to noise ratio (SNR) of both velocity components measured by the fifth sensor (x=2.86 cm). The top row shows contours of the median SNR in cells of 2×2 ${\mathrm{cm}}^{2}$. The bottom row shows the median SNR’s polar contours in cells of 0.02π rad × 2 cm for source states with an *x*-coordinate between x=−7.14 cm and x=12.86 cm, indicating how the movement direction of a dipole φ influences the SNR. Simulated potential flow measurements (Equation (Equation 1)) and the same measurements with additive Gaussian distributed noise values ($\sigma =1.5\times 10^{-5}$ m/s, μ=0 m/s) were used to compute the SNR. Specifically, the SNR was computed as the frequency power ratio between the noisy measurements and noise floor at the source frequency (f=45 Hz). The frequency power was computed by a discrete Fourier transform (DFT) using a Hamming window.

**Figure 4 sensors-21-04558-f004:**
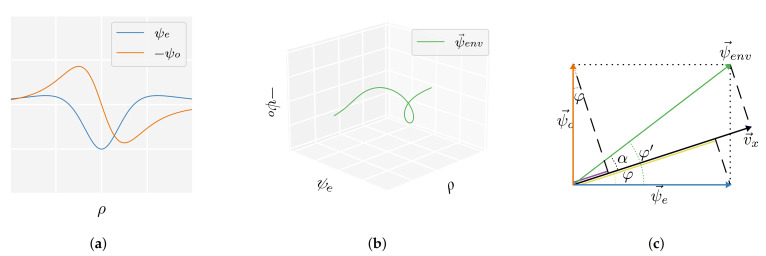
Graphical illustration of the movement direction estimation from the measured velocity and the source’s position ***p*** = 〈*b*, *d*〉 (**a**) A view of *ψ_e_*(*ρ*) and *ψ_o_*(*ρ*) along the sensor array. (**b**) The values of the wavelets can be interpreted as vectors (${\vec{\psi }}_{e}$ and ${\vec{\psi }}_{o}$) in a 3D *ψ*_*e*_–*ψ*_*e*_–*ρ* space. Their vector combination ${\vec{\psi }}_{env}$ = ${\vec{\psi }}_{e}$ + ${\vec{\psi }}_{o}$ is a fixed 3D wavelet structure that can be constructed solely from the source’s previously determined position ***p***. This vector ${\vec{\psi }}_{env}$ has a magnitude $\psi_{env}$ = $\sqrt{\Psi_{e}^{2}+\Psi_{o}^{2}}$ and angle *ψ^’^* = atan *ψ_o_*/*ψ_e_*. The measured velocities—which are linear combinations of *ψ_e_* and *ψ_o_*—can be viewed as a 2D projection of this 3D wavelet. For instance, a projection on the *ρ–ψ_e_* plane (bottom plane) yields *ψ_e_* for *φ* = 0 rad. For a general angle *φ*, the measured velocity profile is a projection on a plane through the *ρ* axis subtending an angle *φ* with the *ρ–φ_e_* plane. (**c**) Diagram illustrating the geometric relation between a measured *v_x_*, the angles *α* and *φ^’^* which are constrained via *Ψ_env_*, and the movement orientation *φ*. We show a slice of ${\vec{\psi }}_{env}$ (green) in the *ψ_e_*–*ψ_o_* plane for a fixed value of *ρ*. The velocity value at this fixed *ρ* is a vector ${\vec{v}}_{x}$ (black) in this space. It has a length *v_x_* ∝ *ψ_e_* cos *φ* + *ψ_o_* sin *φ* and has angle *φ*. The contributions of ${\vec{\psi }}_{e}$ (blue) and ${\vec{\psi }}_{o}$ (orange) to *v_x_* are shown in yellow and purple. The angle *φ^’^* of ~ ${\vec{\psi }}_{env}$ can be computed directly from an estimated source position. Given that the difference between *φ* and *φ^’^* is *α* = acos *v_x_*/*ψ_env_*, we can compute an estimate of *φ* at every sensor using only measured velocity values and a position estimate.

**Figure 5 sensors-21-04558-f005:**
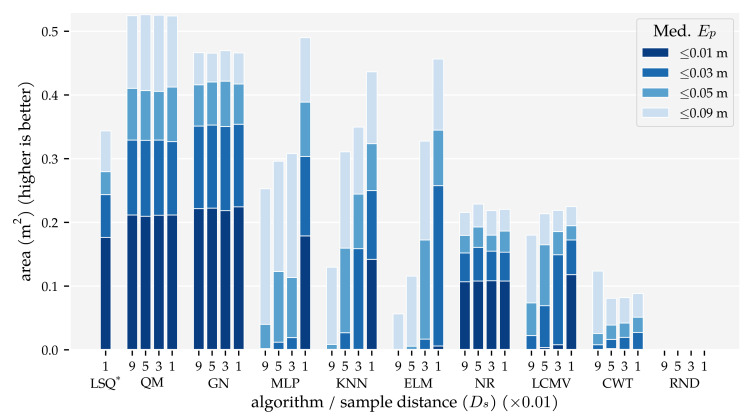
Total areas with a median position error $E_{p}$ (Equation (Equation 4)) below 1 cm, 3 cm, 5 cm, and 9 cm for the training and optimisation sets with a varying minimum distance between source states $D_{s}$ (see Section 2.3 and Table 1) and the (x + y) sensor configuration at the $\sigma =1.0\times 10^{-5}$ m/s noise level. The median position error was computed for 2×2 $cm^{2}$ cells. Note, the bar for LSQ* is based the (x + y) condition in Analysis Method 2.

**Figure 6 sensors-21-04558-f006:**
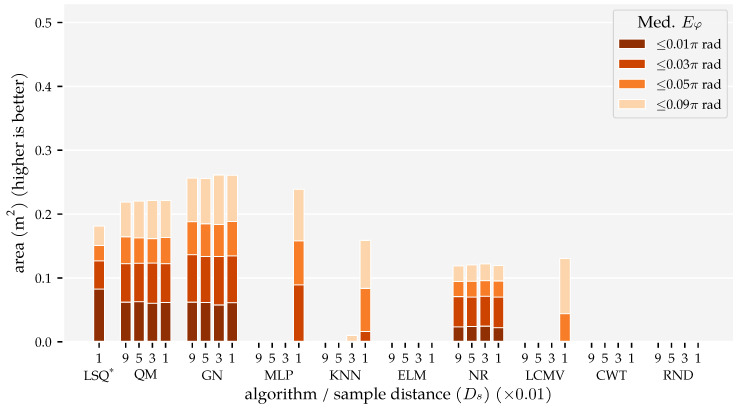
Total areas with a median movement direction error $E_{\varphi }$ (Equation (Equation 5)) below 0.01π rad, 0.03π rad, 0.05π rad for the training and optimisation sets with a varying minimum distance between source states $D_{s}$ (see Section 2.3 and Table 1) and the (x + y) sensor configuration at the $\sigma =1.0\times 10^{-5}$ m/s noise level. The median movement direction error was computed for 2×2 $cm^{2}$ cells. Note, the bar for LSQ* is based on the (x + y) condition in Analysis Method 2.

**Figure 7 sensors-21-04558-f007:**
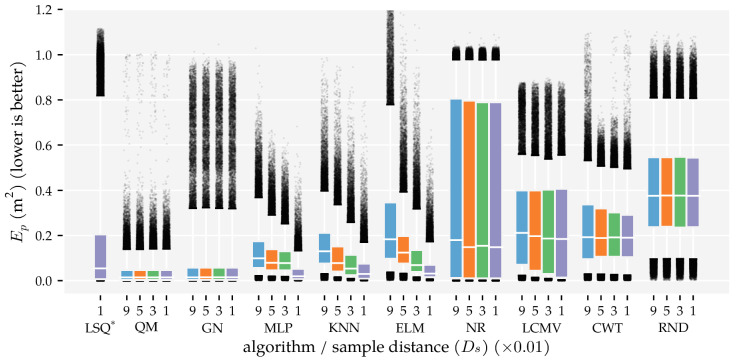
Boxplots of the position error distributions for all dipole localisation algorithms in each condition of first analysis method. This analysis varied the minimum distance $D_{s}$ between source states in the training and optimisation set (see Section 2.3 and Table 1) and used the (x + y) sensor configuration at the $\sigma =1.0\times 10^{-5}$ m/s noise level. The whiskers of the boxplots indicate the 5th and 95th percentiles of the distributions. Predictions with errors outside these percentiles are shown individually. LSQ* is based on the (x + y) condition in Analysis Method 2.

**Figure 8 sensors-21-04558-f008:**
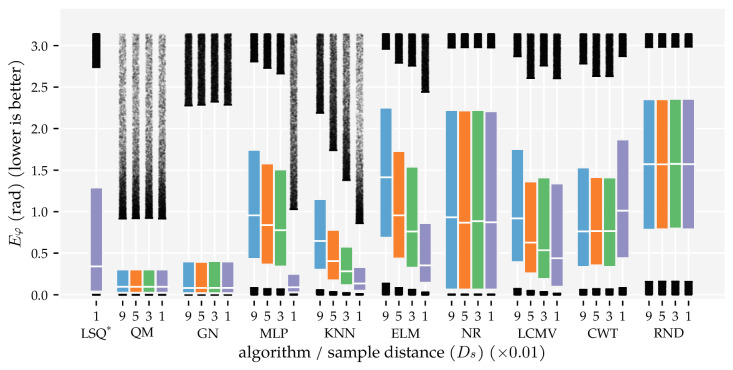
Boxplots of the movement direction error distributions for all dipole localisation algorithms in each condition of first analysis method. This analysis varied the minimum distance $D_{s}$ between source states in the training and optimisation set (see Section 2.3 and Table 1) and used the (x + y) sensor configuration at the $\sigma =1.0\times 10^{-5}$ m/s noise level. The whiskers of the boxplots indicate the 5th and 95th percentiles of the distributions. Predictions with errors outside these percentiles are shown individually. LSQ* is based on the (x + y) condition in Analysis Method 2.

**Figure 9 sensors-21-04558-f009:**
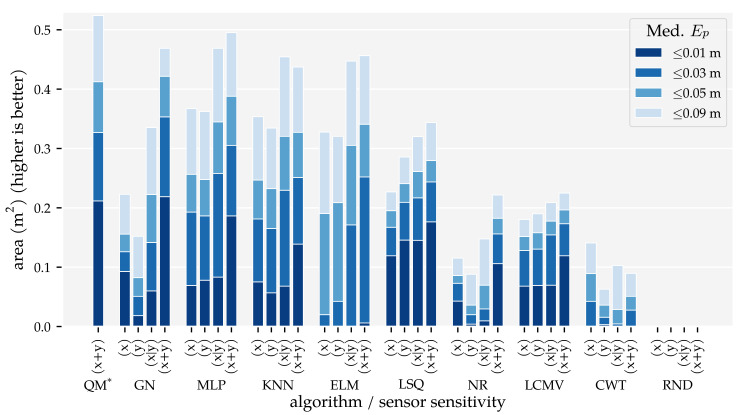
Total area with a median position error $E_{p}$ (Equation (Equation 4)) below 1 cm, 3 cm, 5 cm, and 9 cm for varying sensitivity axes of the sensors: (x + y) measured both velocity components at all sensors, (x|y) alternated measuring $v_{x}$ and $v_{y}$ for subsequent sensors, (x) measured only $v_{x}$ at all sensors, (y) measured only $v_{y}$ at all sensors. This analysis method used the $D_{s}=0.01$ training and optimisation set and the $\sigma =1.0\times 10^{-5}$ m/s noise level. The median position error was computed in 2×2 $cm^{2}$ cells. Note, the bar for QM* is based on the $D_{s}=0.01$ condition in Analysis Method 1.

**Figure 10 sensors-21-04558-f010:**
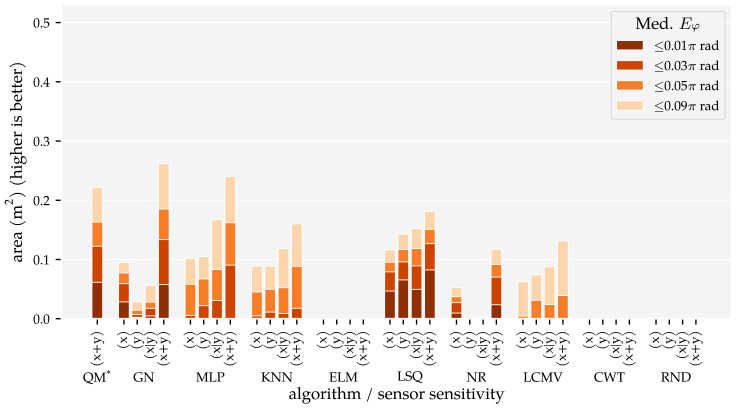
Total area with a median movement direction error $E_{p}$ (Equation (Equation 4)) below 1 cm, 3 cm, 5 cm, and 9 cm for varying sensitivity axes of the sensors: (x + y) measured both velocity components at all sensors, (x|y) alternated measuring $v_{x}$ and $v_{y}$ for subsequent sensors, (x) measured only $v_{x}$ at all sensors, (y) measured only $v_{y}$ at all sensors. This analysis method used the $D_{s}=0.01$ training and optimisation set and the $\sigma =1.0\times 10^{-5}$ m/s noise level. The median movement direction error was computed in 2×2 $cm^{2}$ cells. Note, the bar for QM* is based on the $D_{s}=0.01$ condition in Analysis Method 1.

**Figure 11 sensors-21-04558-f011:**
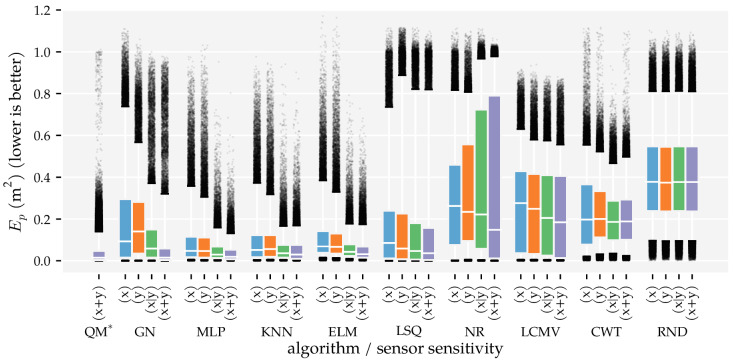
Boxplots of the position error distributions for all algorithms in the second analysis method. This analysis varied the sensitivity axes of the sensors: (x + y) measured both velocity components at all sensors, (x|y) alternated measuring $v_{x}$ and $v_{y}$ for subsequent sensors, (x) measured only $v_{x}$ at all sensors, (y) measured only $v_{y}$ at all sensors. The $D_{s}=0.01$ training and optimisation set and $\sigma =1.0\times 10^{-5}$ m/s noise level were used. The whiskers of the boxplots indicate the 5th and 95th percentiles of the distributions. Predictions with errors outside these percentiles are shown individually. QM* is based on the $D_{s}=0.01$ condition in Analysis Method 1.

**Figure 12 sensors-21-04558-f012:**
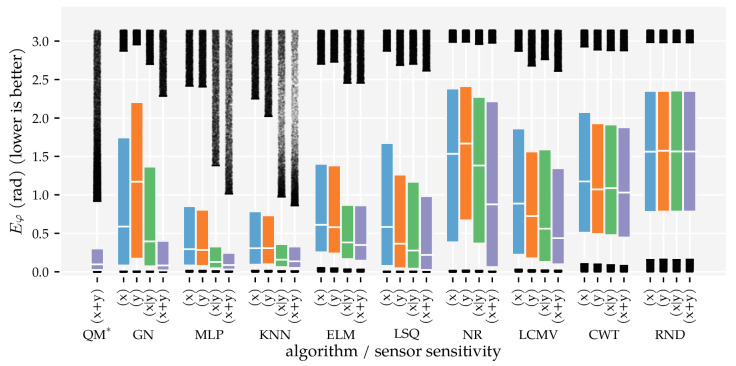
Boxplots of the movement direction error distributions for all algorithms in the second analysis method. This analysis varied the sensitivity axes of the sensors: (x + y) measured both velocity components at all sensors, (x|y) alternated measuring $v_{x}$ and $v_{y}$ for subsequent sensors, (x) measured only $v_{x}$ at all sensors, (y) measured only $v_{y}$ at all sensors. The $D_{s}=0.01$ training and optimisation set and $\sigma =1.0\times 10^{-5}$ m/s noise level were used. The whiskers of the boxplots indicate the 5th and 95th percentiles of the distributions. Predictions with errors outside these percentiles are shown individually. QM* is based on the $D_{s}=0.01$ condition in Analysis Method 1.

**Figure 13 sensors-21-04558-f013:**
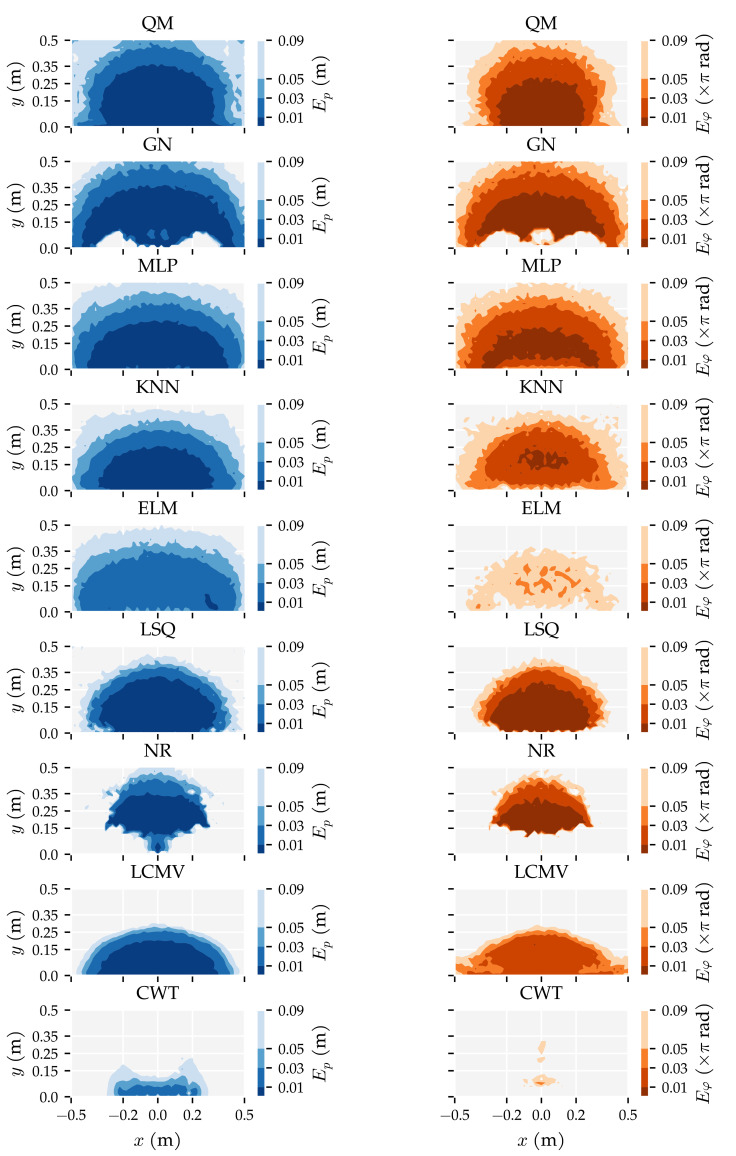
Spatial contours of the median position error $E_{p}$ (blue) (Equation (Equation 4)) and median movement direction error $E_{\varphi }$ (orange) (Equation (Equation 5)) of the predictors using the largest training and optimisation set ($D_{s}=0.01$) and 2D sensitive sensors (x + y) at the $\sigma =1.0\times 10^{-5}$ m/s noise level. The algorithms are ordered with an increasing overall median position error. The median errors were computed in 2×2 $cm^{2}$ cells.

**Figure 14 sensors-21-04558-f014:**
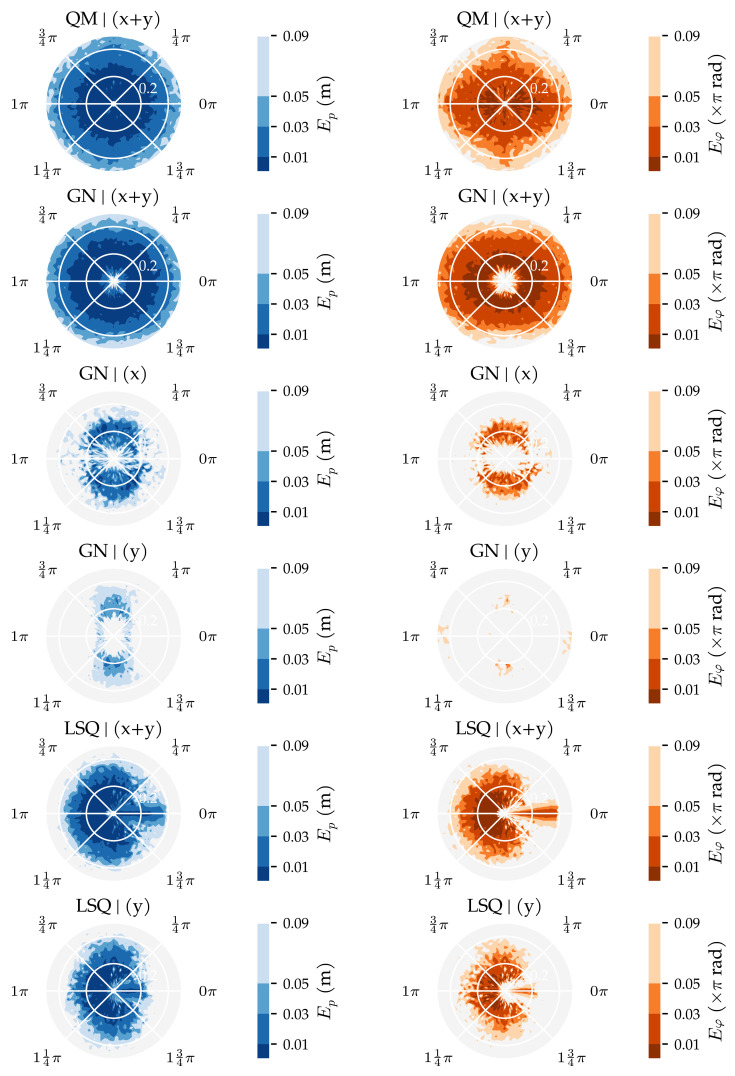
These figures indicate how the movement direction φ and distance *d* of a source state influence the median position error $E_{p}$ (blue) (Equation (Equation 4)) and median movement direction error $E_{\varphi }$ (orange) (Equation (Equation 5)). The quadrature method (QM), Gauss–Newton (GN), and least square curve fit (LSQ) predictors were used with three sensor configurations: (x + y), (x), (y) at the $\sigma =1.0\times 10^{-5}$ m/s noise level. The median errors were computed in cells of 0.01π rad× 1 cm.

**Figure 15 sensors-21-04558-f015:**
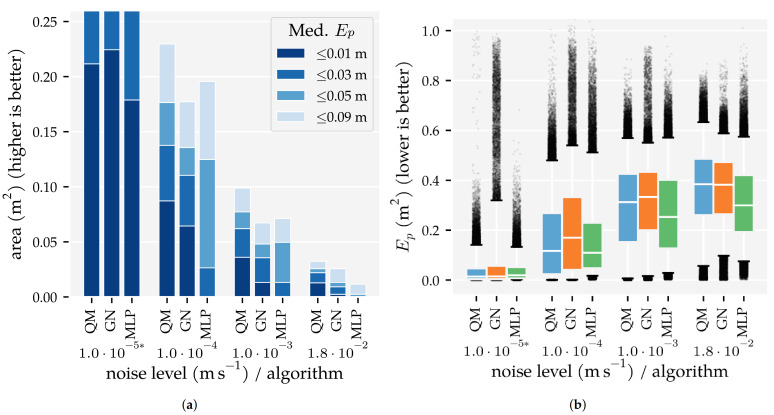
An overview of the position error *E_p_* (Equation (Equation 5)) of QM, GN, and MLP using simulated sensors with higher velocity equivalent noise levels. (**a**) Total areas with a median position error *E_p_* below 1 cm, 3 cm, 5 cm, and 9 cm. (**b**) Boxplots of the position error distributions, whiskers indicate the 5th and 95th percentiles of the distributions. Predictions with errors outside these percentiles are shown individually. The values for σ = 1.0 × 10^‒5^ m s^‒1^ are based on the *D_s_* = 0.01 condition in Analysis Method 1. The (x + y) sensor configuration was used. The MLP was re-trained for each noise level. Both the MLP and GN used the optimal hyperparameter values from the *D_s_* = 0.01 condition of Analysis Method 1.

**Table 1 sensors-21-04558-t001:** An overview of the data sets used in this study. The minimum distance between samples $D_{s}$ controls the number of source states. A source state is specified by a position p and orientation φ. The distance between two states was computed as the Euclidean distance in the combined *x*–*y*–φ/2π space containing all possible combinations of source positions and movement directions. The orientation dimension was divided by 2π to balance the number of positions and orientations. The testing data set has a different number of source states than the training set with $D_{s}=0.01$, due to the randomness of Poisson Disc sampling [37]. The average distance to the closest neighbour within each data set is indicated for both the position and orientation to support the interpretation of $D_{s}$.

Type	Min. Sample Distance ($D_{s}$)	Num. States	Avg. Min $D_{s,p}$(*m*)	Avg. Min $|2\pi D_{s,\varphi }|$ (rad)
training	0.09	169	$2.76\times 10^{-2}$	$1.81\times 10^{-2}$
training	0.05	874	$1.21\times 10^{-2}$	$3.55\times 10^{-3}$
training	0.03	3796	$5.72\times 10^{-3}$	$8.28\times 10^{-4}$
training	0.01	90,435		
testing	0.01	90,502		

**Table 2 sensors-21-04558-t002:** Properties of the dipole localisation algorithms. The ‘Limited to domain’ column indicates whether the algorithm can produce predictions outside the simulated domain (see Figure 2). The ‘Limited to sample’ column indicates whether the algorithm is able to produce a prediction that is not present in the training set.

Algorithm	Type	Training	Hyperparameters	Limited to Domain	Limited to Sample
RND	—	no	no	yes	no
LCMV [12,13]	template-based	yes	no	yes	yes
KNN	template-based	yes	yes	yes	no
CWT [5]	template-based	yes	yes	yes	no
ELM [9,10]	neural network	yes	no	no	no
MLP [7,9]	neural network	yes	no	no	no
GN [8]	model-based	no	yes	yes	no
NR [8]	model-based	no	yes	yes	no
LSQ	model-based	no	yes	yes	no
QM	model-based	no	yes	yes	no

**Table 3 sensors-21-04558-t003:** Training and prediction time measurements of all dipole localisation algorithms. The (x + y) sensor configuration was used combined with the largest training and optimisation set $D_{s}=0.01$.

Algorithm	Avg. Prediction Time	Relative to MLP	Total Training Time
RND	$3.2\times 10^{-4}$ s	0.9	
MLP	$3.6\times 10^{-4}$ s	1.0	12 min 0 s
ELM	$4.3\times 10^{-4}$ s	1.2	1 min 52 s
KNN	$9.7\times 10^{-4}$ s	2.7	
GN	$1.4\times 10^{-3}$ s	3.9	
LSQ	$2.8\times 10^{-3}$ s	7.8	
QM	$3.4\times 10^{-3}$ s	9.6	
LCMV	$1.3\times 10^{-2}$ s	37.1	
NR	$6.3\times 10^{-2}$ s	176.3	
CWT	$1.1\times 10^{-1}$ s	311.1	

## Data Availability

The data generated in this study, including the algorithms’ predictions and the training and test source states, are available from Zenodo [48]. Our implementation of the algorithms in MATLAB R2018a [38] as used in this publication are also available from Zenodo [49].

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
