# Peer review of "The Quadrature Method: A Novel Dipole Localisation Algorithm for Artificial Lateral Lines Compared to State of the Art"

_sensors, 2021, doi:10.3390/s21134558_

Round 1

Reviewer 1 Report

In this study, the authors compare and review 10 localization algorithms for a dipole source using sensors of an "artificial lateral line".  They assess the accuracy of how these algorithms predict the sources' position and movement direction. They also introduce a novel algorithm termed the quadrature method.  By varying the amount of training, optimization of data and sensitivity of the sensors they asses how performance of the algorithms is affected. They find that the area in which algorithms predict position and movement direction does not increase with training or the optimization of data; however, template-based algorithms and neural networks improve performance with training and data optimization. The novel quadrature method provided the best performance with 2D sensors independent of data set size. 
General comments:
The manuscript is generally well written and provides a comprehensive review of the of localization algorithms which will be of interests for the field of hydrodynamic imaging, artificial lateral line design and robotics. 
In sections 3.1 and 3.2 position and movement direction errors were compared between the different algorithms. Would it be possible to assess whether there are statistically significant differences between the methods?
Specific comments:
There are two "Table 1" on page six and page seven
Line 201: This should accordingly refer to Table 2 and the numbering and references to the following tables should be adjusted.
Figures 7-12: It seems the error distribution in some figures is cut off at the top and some legends cover the error distribution. Would it be better to increase the range of the y-axes?

Author Response

We thank the reviewer for reviewing our submitted manuscript. We have addressed all comments in this reply and the suggestions have led to constructive changes to the manuscript. Please see the attachment for the revised manuscript in which changes are indicated in blue.

Point 1: In sections 3.1 and 3.2 position and movement direction errors were compared between the different algorithms. Would it be possible to assess whether there are statistically significant differences between the methods?

Due to the number of datapoints, all differences that we tested are statistically significant. Specifically, we evaluated a repeated measures anova, which found that the position error was statistically significantly different between the algorithms, F(2.74, 248242.2) = 70462.16, p < 0.0001, generalized eta squared = 0.37. Post-hoc analyses with a Bonferroni adjustment revealed that all the pairwise differences are significant at p<0.0001. It should be noted that the distributions are not normally distributed and therefore do not meet the requirements for these tests.

All in all, we do not think that statistical analysis provides valuable additional information to our analysis. We tried to present the results in a way that the differences between the algorithms’ performance can be interpreted without relying on their statistical differences.

Point 2: There are two "Table 1" on page six and page seven. Line 201: This should accordingly refer to Table 2 and the numbering and references to the following tables should be adjusted.

Thank you for spotting this issue. The table numbering has been adjusted in the revised version.

Point 3: It seems the error distribution in some figures is cut off at the top and some legends cover the error distribution. Would it be better to increase the range of the y-axes?

The legends in the box-plots indeed overlap with some of the error distributions. These figures in the revised version now use the x-tick labeling style of the bar-plots instead of a legend. We favoured this over changing the y-range, as that would introduce empty space at the top.

Reviewer 2 Report

The study compares the performance of 10 dipole source localisation algorithms in two-dimensional computer simulations. Nine of these algorithms come from previously published work (from different research groups) and the last one is a novel algorithm proposed by the authors. The experimental methods and results are described clearly in the paper. In principle, I am in favour of publication of the manuscript. Yet, I see few areas of improvement that the authors may wish to consider.  

1) Animals and robots operate in 3D in noisy environments. The problem formulation described in this manuscript (i.e., 2D dipole source localisation) is rather simple and the performance results may not scale proportionally  to 3D dipole source localisation problem. Similarly, the shape and movement of the sensing platform itself (with sensors onboard) would have a significant impact on the flow fields which were omitted in this analysis. I believe discussing these points clearly in the Discussion would help readers interpret the presented results more objectively.

2) Reading the manuscript, one might wonder the complexity and difficulty of 2D dipole source localisation problem. Is there a unique flow field signal for each dipole source location and orientation? If the answer is yes, what makes this problem challenging; signal to noise ratio, spatial resolution (number of sensors)? Detailed analysis of the problem (may be after description of dipole flow fields in the methods) would help the reader appreciate research objectives of the manuscript more.

3) Continuing from point 2, two important parameters (signal to noise ratio and number of sensors) were kept fixed through out the analysis. Extending results on multiple test configuration (lower SNR and lower/higher spatial resolution without changing the length of the ALL) would definitely increase the validity of results. This analysis can be limited to top performing three algorithms at Ds < 0.01.

4) How realistic is it to have sensitivity in both Vx and Vy directions for the same sensor? Please discuss this from both LL and ALL perspectives? 

5) Minor comments.

i) Figure 1 was introduced too early. It is difficult to grasp what figure 1 shows without going trough the equations.

ii) I am having trouble accessing links provided in ref 41 and 42.

iii) The paper is rather long and dense. Is it really necessary to describe how previously published algorithms work in the methods? Can box plots go to supplementary information? Also how important is it to compare performance at different Ds values? It looks like Ds = 0.01 gives best shot to all. In this way, figures 5 and 9 and figures 6 and 10 can be consolidated into one figure each?  Although it is mentioned in the methods, It may be useful to re-mention in captions of the figures related to analysis 1 that both Vx and Vy were available to algorithms.

iv) I think MLP benefitting from larger training dataset is sort of expected and I am not sure how important this is to mention in the abstract.

v) It may be useful to discuss how findings of dipole source localisation may extend to characterising other flow regimes in the light of previous literature (e.g. for instance, localising bluff bodies in Karman vortex streets) or how movement or shape of the sensing platform impact flow sensing. Some useful references:

https://journals.biologists.com/jeb/article/213/22/3819/33461/The-flow-fields-involved-in-hydrodynamic-imaging

https://iopscience.iop.org/article/10.1088/1748-3182/7/3/036004/meta

https://royalsocietypublishing.org/doi/full/10.1098/rsif.2014.0467

https://www.nature.com/articles/ncomms11044?origin=ppub

https://journals.biologists.com/jeb/article/211/18/2950/17664/Swimming-kinematics-and-hydrodynamic-imaging-in

Author Response

We thank the reviewer for reviewing our submitted manuscript. We have addressed all comments in this reply and the suggestions have led to constructive changes to the manuscript. Please see the attachment for the revised manuscript in which changes are indicated in blue.

Point 1: … The problem formulation described in this manuscript (i.e., 2D dipole source localisation) is rather simple and the performance results may not scale proportionally to 3D dipole source localisation problem. … I believe discussing these points clearly in the Discussion would help readers interpret the presented results more objectively.

Thank you for this suggestion, we indeed did not discuss the generalizability of our results to more complex environments (in 3D or with self-motion). These possible extensions to the performed research are indeed relevant to consider. In the revised version, we therefore briefly mentioned these points in the Discussion section. 

Point 2: Detailed analysis of [the complexity and difficulty of 2D dipole source localisation] would help the reader appreciate the research objectives of the manuscript more.

The velocity profiles are indeed unique to each location and orientation. The challenge for the tuned algorithms is the quality of data, which is reflected in the SNR, spatial resolution, and amount of data. In addition, the sensor arrays only capture a part of the velocity profile, which obscures part of the information for the inverse mapping. We added a brief discussion of these points in Section 2.1.

Point 3: two important parameters (signal to noise ratio and number of sensors) were kept fixed throughout the analysis. Extending results on multiple test configurations (lower SNR and lower/higher spatial resolution without changing the length of the ALL) would definitely increase the validity of results.

Whereas a comparison between the algorithms for different Array Densities and Sensor Noise Levels would be interesting, Ds <0.01 might cause ceiling effects to appear quite rapidly. Therefore, we have performed and added a comparison of QM, GN, and the MLP at lower SNRs using the Ds=0.01 training set. This new comparison is described in Section 2.3, the results are presented in Section 3.3, and discussed in Sections 4.3 and 4.4. Several figures are also added to Appendix E for this comparison. We did not perform the full hyperparameter validations because the current results already show a clear trend. 

We refer to Boulogne et al. 2017 for an analysis with multiple spatial resolution.

  • Boulogne, L. H., Wolf, B. J., Wiering, M. A., & van Netten, S. M. (2017). Performance of neural networks for localizing moving objects with an artificial lateral line. Bioinspiration & Biomimetics, 12(5), 56009. https://doi.org/10.1088/1748-3190/aa7fcb

Point 4: How realistic is it to have sensitivity in both Vx and Vy directions for the same sensor? Please discuss this from both LL and ALL perspectives?

There are several 2D sensitive flow sensors in literature which could benefit from algorithms using both velocity components: 

  • Que, R.; Zhu, R.  A Two-Dimensional Flow Sensor with Integrated Micro Thermal Sensing Elements and a Back PropagationNeural Network.Sensors2013,14, 564–574.  doi:10.3390/s140100564.
  • Pjetri, O.; Wiegerink, R.J.; Krijnen, G.J.M. A 2D particle velocity sensor with minimal flow-disturbance.  2015 IEEE SENSORS.IEEE, 2015, pp. 1–4.  doi:10.1109/ICSENS.2015.7370512.
  • Lei, H.; Sharif, M.A.; Tan, X. Dynamics of Omnidirectional IPMC Sensor: Experimental Characterization and Physical Modeling.IEEE/ASME Trans. Mechatronics2016,21, 601–612.  doi:10.1109/TMECH.2015.2468080.
  • Wolf, B.J.; Morton, J.A.S.; MacPherson, W.N.; van Netten, S.M. Bio-inspired all-optical artificial neuromast for 2D flow sensing.Bioinspir. Biomim.2018,13, 026013.  doi:10.1088/1748-3190/aaa786.

In addition, hair cells with varying orientations in close proximity have been observed in the ear of fish (Lu et al. 2001) and on the body of the Xenopus laevis frog (Dijkgraaf 1963).

  • Dijkgraaf, S.  The Functioning and Significance of the Lateral-Line Organs.Biol.  Rev.1963, 38, 51–105.   doi:10.1111/j.1469-185X.1963.tb00654.x.
  • Lu, Z.; Popper, A. Neural response directionality correlates of hair cell orientation in a teleost fish.J. Comp. Physiol. A Sensory,Neural, Behav. Physiol. 2001, 187, 453–465.  doi:10.1007/s003590100218.

We briefly mentioned these points in the Introduction of the revised version.

Point 5.1: Figure 1 was introduced too early.

We agree with your assessment and moved the paragraph explaining Figure 1 to after the definitions of the potential flow in the same subsection. In addition, we changed the caption to avoid mention of rho, which is not yet introduced in that section. Instead the caption now mentions the sensors’ and source’s x-coordinates: x and b. Both of these variables are introduced in that subsection.

Point 5.2:  I am having trouble accessing links provided in ref 41 and 42.

The Zenodo and Github repositories are not publicly accessible pending the publication of the manuscript (only the link to the Github repository should give a 404 error, though). We have uploaded the contents of these repositories to this link https://ufile.io/s1ad6lx6 (password: Hydrodynamical Imaging) which can be accessed anonymously, to facilitate reviewing our work. 

Point 5.3a: The paper is rather long and dense. Is it really necessary to describe how previously published algorithms work in the methods? 

We have tried to reduce our explanation of the algorithms to only include the novel aspects of our application of the algorithms and those details important for the reproducibility of our work. This includes the extension for known algorithms to 2D inputs, which is not trivial in all cases. Moreover, most of the algorithms require at least some fine-tuning in order to perform well. We find it important to mention our decisions in this process. To do that properly we have to explain some of the algorithms in more detail.

Still, we found a couple of points where we could shorten our manuscript:

  1. We moved the tables with the final hyperparameter values to the appendix, which reduces the main body by two pages.
  2. We refer to the tables with hyperparameters only in Section 2.4 instead of for each algorithm individually.
  3. We shortened the descriptions of KNN and CWT. 

Point 5.3b: Can box plots go to supplementary information? 

The boxplots are an integral part of our analysis, especially to compare QM with GN. We prefer to keep them in the main body.

Point 5.3c: Also how important is it to compare performance at different Ds values?

With this analysis we aim to show how much training data some of the algorithms require to approach the performance of optimisation-based algorithms. The comparison shows the potential of the MLP (and KNN) in situations where a potential flow model does not produce good enough results.

Point 5.3d: Although it is mentioned in the methods, It may be useful to re-mention in captions of the figures related to analysis 1 that both Vx and Vy were available to algorithms. 

We have updated all figure captions to mention which sensor configuration, training-optimisation set, and noise-level were used.

Point 5.4: I think MLP benefitting from larger training dataset is sort of expected and I am not sure how important this is to mention in the abstract.

This segment of the abstract aims to convey that the MLP can only (barely) match the performance with a density of measurements (every ~2 cm's) that is impractical in most applications. We have re-written it to better match the intended message.

Point 5.5: It may be useful to discuss how findings of dipole source localisation may extend to characterising other flow regimes in the light of previous literature (e.g. for instance, localising bluff bodies in Karman vortex streets) or how movement or shape of the sensing platform impact flow sensing.

Although those applications are out of the scope of our work, we briefly mention the influence of shape and self-motion in the discussion section.

Reviewer 3 Report

The paper compares various localization algorithms for a dipole based on ALLs in three aspects and puts forward a new method. The review is systematic and enlightening. The following comments shall be helpful for improving the paper.

At the end of Section 1, it will be better to present the structure of the article so that readers can have a general idea of it.

Your contribution, not only limiting to comparing various methods, could be highlighted at the end of Introduction.

There is a mistake in the number of tables that ‘figure 1’ appear twice.

The details about the QM could be put in the main body to show your contribution and innovation.

Why the QM, not other methods, is chosen as the baseline should be discussed further.

Author Response

We thank the reviewer for reviewing our submitted manuscript. We have addressed all comments in this reply and the suggestions have led to constructive changes to the manuscript. Please see the attachment for the revised manuscript in which changes are indicated in blue.

Point 1: At the end of Section 1, it will be better to present the structure of the article so that readers can have a general idea of it.

In the revised version, we added a paragraph at the end of the Introduction, outlining the structure of the paper.

Point 2: Your contribution, not only limiting to comparing various methods, could be highlighted at the end of Introduction.

The last paragraph of the introduction was intended to show our contributions. It indeed focuses mostly on comparing the algorithms. In the revised version, we elaborated on QM as our contribution as well.

Point 3: There is a mistake in the number of tables that ‘figure 1’ appear twice.

Thank you for spotting this issue. The table numbering has been fixed in the revised version.

Point 4: The details about the QM could be put in the main body to show your contribution and innovation.

While we agree that the QM and its details are a large part of the contribution of our work, we prefer to keep those details in the appendix. In line with reviewer 2, we believe that the main-body of the manuscript is rather long. Therefore, we opted to reduce the methods section to contain only the information necessary to reproduce our work. For the QM, we explain how the algorithm works in the methods section but prefer to leave the derivations in the appendix.

Point 5: Why the QM, not other methods, is chosen as the baseline should be discussed further.

We used the random predictor and the off-the-shelf least squares curve fit (LSQ) algorithms as baselines in our comparisons. The QM came out of those comparisons as the best performing algorithm.

We believe the sentence “We benchmark them against our novel quadrature method…” in the abstract may have confused our intended message. Therefore, that sentence has been changed to: “We compare them with our novel quadrature method…”.
